# Square$\chi$PO: Differentially Private and Robust $\chi^2$-Preference Optimization in Offline Direct Alignment

Xingyu Zhou [1]   Yulian Wu [2]   Wenqian Weng [1]   Francesco Orabona [2]

## Abstract

In this paper, we theoretically study the offline alignment of language models with human preference feedback, under both preference label corruption and privacy protections. To this end, we propose Square$\chi$PO, a simple one-line change to $\chi$PO where the standard log-loss is replaced by a new square loss over probability. Thanks to the inherent properties of this new loss, we have advanced the state-of-the-art of differentially private and robust offline direct alignment. Specifically, for the local model of label privacy, Square$\chi$PO is the first algorithm that attains an optimal rate based on single-policy concentrability even with general function approximations. It also gives the first result under the central model of privacy protection over both prompts (responses) and labels. On the robustness side against Huber label corruption, Square$\chi$PO is the first alignment method that has a meaningful theoretical guarantee under general function approximations. More importantly, Square$\chi$PO can address privacy protection and corruption *simultaneously*, where an interesting separation is observed, implying that the order of privacy and corruption matters. Furthermore, we show that Square$\chi$PO can also be easily extended to handle the scenario of the general preference model with state-of-the-art guarantees under corruption and privacy. Last but not least, all of our theoretical guarantees enjoy a unified analysis, building upon a new result on the generalization error bounds of least-square regression under corruption and privacy constraints, which we believe is of independent interest to the community.

[1]Wayne State University, USA [2]King Abdullah University of Science and Technology, Saudi Arabia. Correspondence to: Xingyu Zhou <xingyu.zhou@wayne.edu>.

*Proceedings of the 42$^{nd}$ International Conference on Machine Learning*, Vancouver, Canada. PMLR 267, 2025. Copyright 2025 by the author(s).

## 1. Introduction

Aligning large language models (LLMs) to human values is crucial for their responsible deployment. Two primary paradigms have emerged: *indirect alignment*, where a reward model is learned before the policy optimized via Reinforcement Learning (RL) (Christiano et al., 2017; Ouyang et al., 2022), and *direct alignment*, an RL-free approach leveraging reparametrization techniques like Direct Preference Optimization (DPO) (Rafailov et al., 2023). Very recently, a variant of DPO, called $\chi$PO (Huang et al., 2024), addresses the overoptimization issue in direct alignment by relying on a significantly weaker condition – single-policy concentrability – making it the first offline direct alignment method with such a guarantee.

Meanwhile, privacy and robustness concerns in the preference datasets of the alignment process have gained significant attention. Membership inference attacks expose privacy vulnerabilities (Feng et al., 2024), while data poisoning undermines label integrity (Casper et al., 2023). Recent efforts have addressed these challenges separately, providing theoretical guarantees for privacy or robustness. On the privacy side, existing theoretical work has primarily focused on simple linear function approximations (Zhou et al., 2025; Chowdhury et al., 2024b; Korkmaz & Brown-Cohen, 2024), which are insufficient for practical scenarios involving non-linear reward or policy function classes (e.g., neural networks).

> **Q1.** For general function approximations, can we achieve optimal (or better) rates under privacy constraints?

**Contribution 1.** We answer **Q1** affirmatively by introducing Square$\chi$PO, a simple variant of $\chi$PO which replaces the log loss with a new square loss over probabilities. For preference label privacy under the local model of Differential Privacy (DP) (Kasiviswanathan et al., 2011; Chaudhuri & Hsu, 2011), Square$\chi$PO achieves the optimal privacy cost, even with general function approximations. Furthermore, under the standard central DP model (Dwork et al., 2006), it provides the first *pure DP* guarantees for the case of general function approximations.

Moving now to the robustness side, Mandal et al. (2024) takes an indirect approach, focusing on the linear setting,

while Chowdhury et al. (2024a) follows a DPO-style direct method, which, however only achieves a suboptimal rate for the linear case and suffers from a non-vanishing suboptimality gap for general function approximations.

> **Q2.** Can we improve these results under label corruption, even for general function approximations?

**Contribution 2.** Our Square$\chi$PO provides an affirmative answer to **Q2**. Specifically, it not only preserves the favorable single-policy concentrability property of $\chi$PO, but also achieves the optimal $\mathcal{O}(1/\sqrt{n})$ rate for general function approximations under the same random-flipping corruption setting as in Chowdhury et al. (2024a). Furthermore, due to the inherent boundedness of our new loss, Square$\chi$PO is the first alignment method to provide meaningful guarantees under stronger Huber label corruption (Huber, 1964), matching the best-known results in the non-preference feedback offline RL setting (Zhang et al., 2022).

Instead of studying privacy protection and robustness to corruption separately, there is growing interest in understanding their interplay, driven by both practical scenarios and theoretical insights, for example, in bandits (Zhou & Zhang, 2024; Wu et al., 2024b; Charisopoulos et al., 2023) or general statistical tasks; please refer to Kamath (2024) for a wonderful recent survey.

> **Q3.** Can we achieve privacy protection and robustness simultaneously, and what are the interplays between them?

**Contribution 3.** Our Square$\chi$PO simultaneously addresses privacy and robustness in offline direct alignment, uncovering interesting interplays between the two. For the local model of label privacy, we examine two settings that differ in the order of privacy and corruption. Square$\chi$PO is adaptive, as it does not require prior knowledge of the specific setting while providing sharp rates. Notably, our results reveal that corruption following privacy leads to worse bounds. For the central model of DP, our findings illustrate that the effects of privacy and corruption are only *additive*. Both are consistent with prior observations in mean estimation and bandits (Zhou & Zhang, 2024; Wu et al., 2024b).

All the above results (including those prior work) are established under the assumption of the Bradley-Terry (BT) preference model (Bradley & Terry, 1952), which implicitly assumes transitive preferences (i.e., $a \succ b, b \succ c \Rightarrow a \succ c$). However, transitivity does not always hold in practice. Building on recent work in the non-private and non-corrupted setting, where general preference models have been explored (Munos et al., 2023; Swamy et al., 2024), it is natural to pose the next question:

> **Q4.** For a general preference model, can we still achieve privacy protection and robustness simultaneously?

**Contribution 4.** We answer this question affirmatively by demonstrating that an iterative version of Square$\chi$PO provides the first set of results for private and robust alignment under a general preference model, achieving guarantees analogous to the results of iterative $\chi$PO (Huang et al., 2024)

Finally, on the technical side, it is often desirable to have a clean and unified analysis across different settings, which in our case includes privacy (local or central models), corruption, as well as BT and general preference models.

> **Q5.** Can we have a unified analysis of Square$\chi$PO?

**Contribution 5.** We answer this question affirmatively by establishing all of our theoretical results through a key common analytical tool: new generalization error bounds for least-square regression under privacy constraints and corruption. Given the widespread use of least-square regression oracles in RL (Agarwal et al., 2019), we believe these results could be of independent interest.

In the interest of space, we relegate the discussion on further related work to Appendix A.

## 2. Preliminaries

### 2.1. Offline Alignment

In the offline alignment problem, there exists a pre-collected preference dataset $\mathcal{D}_{\mathsf{pref}} = \{(x_i, a_i^0, a_i^1, y_i)\}_{i=1}^n$, where each context/prompt $x_i$ is i.i.d. sampled from a distribution $\rho$, and two responses $a_i^0$ and $a_i^1$ are i.i.d sampled from a reference policy $\pi_{\mathsf{ref}}$, i.e., $a_i^0 \sim \pi_{\mathsf{ref}}(\cdot \mid x_i)$ and $a_i^1 \sim \pi_{\mathsf{ref}}(\cdot \mid x_i)$, and finally the preference label $y_i \in \{0, 1\}$ is generated according to some probability distribution, i.e., $y_i \sim \mathrm{Ber}(\mathcal{P}^\star(a_i^1 \succ a_i^0 \mid x_i))$, where $\mathcal{P}^\star(a_i^1 \succ a_i^0 \mid x_i) \in [0, 1]$ is the probability that given $x_i$, $a_i^1$ is preferred over $a_i^0$ and $\mathrm{Ber}(\cdot)$ denotes a Bernoulli distribution. Without loss of generality, we assume that $\rho(x) > 0$ for all $x$ and $\pi_{\mathsf{ref}}(a \mid x) > 0$ for all $x$ and $a$. Depending on the modeling assumption of the preference probability $\mathcal{P}^\star(a_i^1 \succ a_i^0 \mid x_i)$, the (offline) alignment is often categorized into the following two settings.

**Bradley-Terry (BT) preference model (Bradley & Terry, 1952).** In this setting, there exists an unknown true reward function $r^\star : \mathcal{X} \times \mathcal{A} \to [0, R_{\max}]$ that induces the preference probability as follows

$$\mathcal{P}^\star(a_i^1 \succ a_i^0 \mid x_i) = \frac{\exp(r^\star(x_i, a_i^1))}{\exp(r^\star(x_i, a_i^1)) + \exp(r^\star(x_i, a_i^0))}.$$

With the preference dataset $\mathcal{D}_{\mathsf{pref}}$, the goal under this setting is to learn a policy $\widehat{\pi}$ that minimizes the suboptimality gap:

$$\mathsf{SG}(\widehat{\pi}; \pi^\star) := J(\pi^\star) - J(\widehat{\pi}), \tag{1}$$

where $J(\pi) := \mathbb{E}_{x \sim \rho, a \sim \pi(\cdot \mid x)}[r^\star(x, a)]$ and $\pi^\star$ is any comparator policy (e.g., it could be the optimal policy maximiz-

ing $J(\pi)$ or any other policy). For notation simplicity, we will abbreviate $\mathbb{E}_\pi[\cdot] := \mathbb{E}_{x \sim \rho, a \sim \pi(\cdot|x)}[\cdot]$.

**General preference model (Munos et al., 2023).** In this setting, one directly works with a general preference model $\mathcal{P}^\star(a_i^1 \succ a_i^0 \mid x_i)$ without the parametrization of a reward function as above. This general preference model has several advantages over the BT-preference model, e.g., it is better at capturing non-transitive preferences ($a \succ b$, $b \succ c$, $c \succ a$). Without the reward function, the solution concept now becomes *minimax winner* (*von Neumann winner*) (Munos et al., 2023; Swamy et al., 2024; Wang et al., 2023b), which is given by

$$\pi_{\mathsf{MW}} := \operatorname*{argmax}_{\pi \in \Pi} \min_{\pi' \in \Pi} \ \mathcal{P}^\star(\pi \succ \pi'),$$

where $\mathcal{P}^\star(\pi \succ \pi') := \mathbb{E}_{x \sim \rho}[\mathcal{P}^\star(\pi(x) \succ \pi'(x) \mid x)]$ for a pair of policies $\pi, \pi'$ in a policy class $\Pi$. It is often more convenient to work with a scaled and shifted version of $\mathcal{P}^\star(a^1 \succ a^0 \mid x)$ as $\ell^\star(x, a^1, a^0) := 2\mathcal{P}^\star(a^1 \succ a^0 \mid x) - 1$, which leads to an equivalent definition of minimax winner

$$\pi_{\mathsf{MW}} := \operatorname*{argmax}_{\pi \in \Pi} \min_{\pi' \in \Pi} \ \ell^\star(\pi, \pi'), \tag{2}$$

where $\ell^\star(\pi, \pi') := \mathbb{E}_{x \sim \rho, a^1 \sim \pi(\cdot|x), a^0 \sim \pi'(\cdot|x)}[\ell^\star(x, a^1, a^0)]$. Since this minimax winner can be viewed as a *Nash equilibrium* of two-player constant-sum game, our goal in this setting is to minimize the duality gap

$$\mathsf{DG}(\widehat{\pi}) := \max_{\pi \in \Pi} \ell^\star(\pi, \widehat{\pi}) - \min_{\pi \in \Pi} \ell^\star(\widehat{\pi}, \pi).$$

## 2.2. DPO and $\chi$PO

**DPO.** One of the most popular offline alignment algorithms is Direct Preference Optimization (DPO) (Rafailov et al., 2023). Its popularity could be partially attributed to its success in eliminating the reward model learning process, achieved by a reparameterization of reward by the optimal policy of a KL-regularized optimization objective. In particular, under the BT-preference model, given a preference dataset $\mathcal{D}_{\mathsf{pref}}$ and a user-specified policy class $\Pi$, DPO solves

$$\widehat{\pi}_{\mathsf{DPO}} = \operatorname*{argmax}_{\pi \in \Pi} \sum_{(x, a_+, a_-) \in \mathcal{D}_{\mathsf{pref}}} \log[\sigma(\beta h_{\mathsf{DPO}}(x, a_+, a_-))],$$

where $h_{\mathsf{DPO}}(x, a_+, a_-) := \log \frac{\pi(a_+|x)}{\pi_{\mathrm{ref}}(a_+|x)} - \log \frac{\pi(a_-|x)}{\pi_{\mathrm{ref}}(a_-|x)}$, $\sigma(z) = \frac{1}{1+e^{-z}}$ is the sigmoid function, and $\beta > 0$ is some regularization parameter. Here, for any data point $(x, a^0, a^1, y)$ in $\mathcal{D}_{\mathsf{pref}}$, we set $a_+ = a^y$ (the preferred one) and $a_- = a^{1-y}$ (the non-preferred one).

$\chi$**PO.** To address the inherent overoptimization issue in DPO, Huang et al. (2024) recently proposed a simple variant of DPO by introducing an additional $\chi^2$-regularization term,

which leads to the following optimization[1]

$$\widehat{\pi}_{\chi\mathsf{PO}} = \operatorname*{argmax}_{\pi \in \Pi} \sum_{(x, a_+, a_-) \in \mathcal{D}_{\mathsf{pref}}} \log[\sigma(\beta h_{\chi\mathsf{PO}}(x, a_+, a_-))],$$

where $h_{\chi\mathsf{PO}}(x, a_+, a_-) := \phi\left(\frac{\pi(a_+|x)}{\pi_{\mathrm{ref}}(a_+|x)}\right) - \phi\left(\frac{\pi(a_-|x)}{\pi_{\mathrm{ref}}(a_-|x)}\right)$ and $\phi(u) := u + \log u$. Compared to DPO, there is an additional linear term in $\phi(z)$ that introduces *pessimism* (Jin et al., 2021b), which enables a suboptimality gap that only depends on *single policy concentrability* (Rashidinejad et al., 2021). On the other hand, DPO could only achieve a suboptimality gap in terms of *all-policy concentrability coefficient* (Chen & Jiang, 2019) due to the lack of pessimism. Moreover, $\chi$PO can also be extended to handle the general preference model with a meaningful upper bound on the duality gap. Given the stronger performance of $\chi$PO, we will mainly focus on it when we consider robustness and privacy in offline alignment, as discussed below.

## 2.3. Robustness and Privacy in Preference Data

**Label corruption.** In practice, the preference label $y_i$ may not be sampled from the clean distribution $\mathrm{Ber}(\mathcal{P}^\star(a_i^1 \succ a_i^0 \mid x_i))$. To characterize this, we borrow the classic *Huber corruption* model from robust statistics.

**Definition 2.1** ($\alpha$-Huber corruption (Huber, 1964)). We consider the following $\alpha$-Huber corruption: each label is independently sampled from $(1 - \alpha)G_i + \alpha B_i$, where $G_i$ is the clean distribution $\mathrm{Ber}(\mathcal{P}^\star(a_i^1 \succ a_i^0 \mid x_i))$ and $B_i$ is some arbitrary unknown Bernoulli distribution. That is, with probability $\alpha \in [0, 1/2]$, each label is sampled from some bad distribution.

**Label privacy in the local model.** The preference label is often collected via human feedback, which could potentially reveal each person's private information, as discussed before. To this end, a strong privacy protection is to ensure *Local Differential Privacy* (LDP) via a local randomizer. Given the binary data of the preference label, it is natural to consider the classic *randomized response* mechanism.

**Definition 2.2** (Randomized response and $\varepsilon$-LDP (Warner, 1965)). Let $\varepsilon > 0$ be the privacy parameter and $y \in \{0, 1\}$ be the true label. The randomized response (RR) mechanism $\mathcal{R}$ flips $y$ and outputs private $\widetilde{y}$ based on the following distribution

$$\mathbb{P}[\widetilde{y} = y] = \frac{e^\varepsilon}{1 + e^\varepsilon} \text{ and } \mathbb{P}[\widetilde{y} \neq y] = \frac{1}{1 + e^\varepsilon}. \tag{3}$$

This can be easily shown to satisfy $\varepsilon$-LDP, i.e., for any $y, y'$ and any subset $S$ in the range of $\mathcal{R}$ such that

$$\mathbb{P}[\mathcal{R}(y) \in S] \leq e^\varepsilon \cdot \mathbb{P}[\mathcal{R}(y') \in S].$$

---

[1] We ignore the clipping operation for the ease of presentation.

**Interplay between corruption and LDP.** In practice, corruption and LDP protection can exist together, which motivates us to consider their interplay in the following settings.

**Definition 2.3** (CTL and LTC)**.** Given a raw preference dataset $\mathcal{D}_{\text{pref}} = \{(x_i, a_i^0, a_i^1, y_i)\}_{i=1}^n$ and two parameters $\alpha \in [0, 1/2]$, $\varepsilon > 0$, we consider the following two settings that differ in the order of corruption and label privacy protection in the local model:

**Corruption-then-LDP** (CTL)**.** The raw label $y_i$ is first corrupted by the $\alpha$-Huber model, which is then further privatized by $\varepsilon$-LDP RR mechanism, leading to the final preference dataset given by $\widetilde{\mathcal{D}}_{\text{pref}} = \{(x_i, a_i^0, a_i^1, z_i)\}_{i=1}^n$.

**LDP-then-Corruption** (LTC)**.** The raw label $y_i$ is first privatized by $\varepsilon$-LDP RR mechanism, which is then further corrupted by the $\alpha$-Huber model, leading to the final preference dataset given by $\widetilde{\mathcal{D}}_{\text{pref}} = \{(x_i, a_i^0, a_i^1, z_i)\}_{i=1}^n$.

One of our goals is to study whether there exists a separation between the two settings, implying the order of corruption and LDP matters.

*Remark* 2.4. The two settings naturally include corruption-only and privacy-only as special cases by setting $\varepsilon = \infty$ and $\alpha = 0$, respectively. Moreover, it is easy to see that, combining the results of CTL and LTC directly gives us the result for an even practical setting where corruption happens both before and after LDP.

**Differential privacy in the central model.** We will also consider the standard DP definition, which is defined in the central model where the learner has access to the raw data and needs to ensure a similar output on two neighboring datasets.

**Definition 2.5** (($\varepsilon, \delta$)-DP (Dwork et al., 2006))**.** Let $\varepsilon > 0$ and $\delta \in [0, 1]$, and $\mathcal{A}$ be a given offline alignment algorithm. We say $\mathcal{A}$ satisfies $\varepsilon$-DP if for any measurable set $S$ in the range of $\mathcal{A}$

$$\mathbb{P}[\mathcal{A}(\mathcal{D}_{\text{pref}}) \in S] \le e^\varepsilon \cdot \mathbb{P}[\mathcal{A}(\mathcal{D}'_{\text{pref}}) \in S] + \delta,$$

holds for any pair of $(\mathcal{D}_{\text{pref}}, \mathcal{D}'_{\text{pref}})$ that only differs in one sample $(x_i, a_i^0, a_i^1, y_i)$ for some $i \in [n]$. If $\delta = 0$, we simply write $\varepsilon$-DP (i.e., pure DP).

Here, we not only protect the preference label, but also the prompt and responses. As before, we would also like to study the interplay between corruption the central DP. In contrast to the local model, here the label corruption can only happen before the central privacy protection.

**Definition 2.6** (Corruption and DP (cDP))**.** Given a raw preference dataset $\mathcal{D}_{\text{pref}} = \{(x_i, a_i^0, a_i^1, y_i)\}_{i=1}^n$ and two parameters $\alpha \in [0, 1/2]$, $\varepsilon > 0$, we consider the following interplay: each label $y_i$ is first corrupted by $\alpha$-Huber model, resulting in $\bar{\mathcal{D}}_{\text{pref}} = \{(x_i, a_i^0, a_i^1, \bar{y}_i)\}_{i=1}^n$. Then, the learner employs an algorithm $\mathcal{A}$ that is $\varepsilon$-DP with respect to $\bar{\mathcal{D}}_{\text{pref}}$.

---

**Algorithm 1** Square$\chi$PO for CTL and LTC

1: **Input:** Locally private and corrupted preference dataset $\widetilde{\mathcal{D}}_{\text{pref}} = \{(x_i, a_i^0, a_i^1, z_i)\}_{i=1}^n$ under CTL and LTC, privacy parameter $\varepsilon > 0$, regularization coefficient $\beta > 0$, reference policy $\pi_{\text{ref}}$

2: Define

$$\phi(u) := u + \log u \tag{4}$$

$$h_{\chi\text{PO},i} := \phi\left(\frac{\pi(a_i^1 \mid x_i)}{\pi_{\text{ref}}(a_i^1 \mid x_i)}\right) - \phi\left(\frac{\pi(a_i^0 \mid x_i)}{\pi_{\text{ref}}(a_i^0 \mid x_i)}\right) \tag{5}$$

3: Optimize the following objective:

$$\hat{\pi} \leftarrow \underset{\pi \in \Pi}{\arg\min} \sum_{i \in [n]} \left[2\sigma\left(\text{clip}_{2R_{\max}}[\beta h_{\chi\text{PO},i}]\right) - 1 - c(\varepsilon)\bar{z}_i\right]^2,$$

where $c(\varepsilon) := \frac{e^\varepsilon + 1}{e^\varepsilon - 1}$ and $\bar{z}_i = 2z_i - 1$

4: **Output:** $\hat{\pi}$

---

*Remark* 2.7. As before, cDP recovers privacy-only and corruption-only settings by setting $\alpha = 0$ and $\varepsilon = \infty$, respectively.

## 3. Bradley-Terry Preference Model

In this section, we study offline alignment in the BT-preference model under privacy constraints and corruption. We first focus on the interplay between corruption and the label LDP (i.e., CTL and LTC) and then turn to the setting of central DP, i.e., cDP.

### 3.1. Local Model

Our proposed algorithm, Square$\chi$PO in Algorithm 1, is the same for both CTL and LTC, i.e., adaptive. The key modification compared with $\chi$PO is to use a square loss instead of the log loss, plus an additional $c(\varepsilon)$ factor for the private case. We will dive into the intuition about the choice of our loss function in the sequel. Before that, we remark that the clipping $\text{clip}_R(u) = \max\{\min\{u, R\}, -R\}$ with $R = 2R_{\max}$ is adopted in $\chi$PO as well, mainly used for a slightly tighter theoretical bound.

#### 3.1.1. INTUITION BEHIND SQUARE$\chi$PO

We now discuss our new loss function in Square$\chi$PO, highlighting the intuition on how it helps to handle corruption and privacy protection. It is worth noting that our new loss function could be of its own interest even in the standard scenario, i.e., non-private and non-corrupted cases, with DPO-type (rather than $\chi$PO-type) reparameterization.

**1. Square loss over probability.** Without privacy protection

($c(\varepsilon) = 1$), our new loss function essentially reduces to

$$\sum_{i \in [n]} (p_i(\pi) - z_i)^2, \tag{6}$$

where we define $p_i(\pi) := \sigma\left(\text{clip}_{2R_{\max}}[\beta h_{\chi\text{PO},i}]\right)$, while DPO and $\chi$PO essentially adopts the standard log-loss, i.e.,

$$-z_i \log p_i(\pi) - (1 - z_i) \log(1 - p_i(\pi)). \tag{7}$$

In fact, the loss in (6) is often referred to as *Brier score* (Brier, 1950) in probabilistic predictions. One direct observation here is that the Brier score is always upper bounded by 1 while the log-loss can be unbounded, which implies that label corruption under log-loss may have a larger impact than that under the Brier score.

**2. Converting to $\pm 1$ with $c(\varepsilon)$ scaling.** Instead of working with $z_i \in \{0, 1\}$, we convert it to $\bar{z}_i = 2z_i - 1 \in \{1, -1\}$ and we similarly update the probability part. There are two main reasons for this: (i) From (3) of RR, we can easily see that the private mean (under $\pm 1$) is $1/c(\varepsilon)$ of the true mean (probability). This implies that the $c(\varepsilon)$ factor in front of the private data leads to an unbiased estimate of the true probability, which essentially follows from the same intuition as in private mean estimation under RR, since the empirical average mean estimator can also be written as the solution to a square loss; (ii) Recall that for the general preference model, it often works with $\pm 1$ (cf. (2)). As we will see later, this conversion allows us to essentially employ the same technique to analyze both BT-preference and general preference models, altogether.

*Remark* 3.1. We mention in passing that many alignment algorithms draw inspiration from binary classification for their loss functions, in the non-private non-corrupted cases. For instance, in addition to log-loss in DPO and $\chi$PO, SLiC (Zhao et al., 2023) leverages the hinge loss while IPO (Azar et al., 2024) adopts the standard square loss. The key conceptual difference between our square loss and that of IPO lies in the fact that the latter takes the square over the raw log-ratio (i.e., implicit reward) while ours is a square over probability (i.e., an additional sigmoid step is applied). More recently, Tang et al. (2024) proposed a family of loss functions for alignment based on standard supervised learning, including *exponential loss*, *truncated quadratic loss*, and *savage loss*. To the best of our knowledge, our Square$\chi$PO is the first one that proposes to use the Brier score as the loss. In the next section, we will demonstrate its strong theoretical guarantees.

### 3.1.2. THEORETICAL GUARANTEES

In this section, our aim is to establish the suboptimality gap (cf. (1)) of Square$\chi$PO (Algorithm 1), under both CTL and LTC, without knowledge of the setting in advance.

We start with the same assumptions as in $\chi$PO (Huang et al., 2024), i.e., policy realizability and bounded range.

**Assumption 3.2** (Policy realizability). Fix $\beta > 0$. The policy class $\Pi$ satisfies $\pi_\beta^\star \in \Pi$, where $\pi_\beta^\star$ is the optimal policy of the following mixed $\chi^2$-regularized objective:

$$J_\beta^{\chi_{\text{mix}}}(\pi) := \mathbb{E}_\pi[r^\star(x, a)] - \beta \cdot [D_{\chi^2}(\pi \| \pi_{\text{ref}}) + D_{\text{KL}}(\pi \| \pi_{\text{ref}})].$$

The $J_\beta^{\chi_{\text{mix}}}(\pi)$ in $\chi$PO mixes $\chi^2$-regularization with the standard KL-regularization in DPO, which in turn leads to the new reward reparameterization using optimal solution $\pi_\beta^\star$:

$$r^\star(x, a) = \beta\phi\left(\frac{\pi_\beta^\star(a|x)}{\pi_{\text{ref}}(a|x)}\right) + Z_{\beta, r^\star}(x),$$

where we recall that $\phi(u) = u + \log u$ and $Z_{\beta, r^\star}(x)$ is some action-independent normalization term. Thus, Assumption 3.2 essentially implies the implicit reward realizability under the above parameterization.

As in $\chi$PO (Huang et al., 2024), the next assumption asserts that the *implicit reward difference* under any policy in $\Pi$ is upper bounded by some constant.

**Assumption 3.3** (Bounded implicit reward difference). For a parameter $V_{\max} \geqslant R_{\max}$, it holds that for all $\pi \in \Pi$, $x \in \mathcal{X}$, and $a, b \in \mathcal{A}$,

$$\left|\beta\phi\left(\frac{\pi(a \mid x)}{\pi_{\text{ref}}(a \mid x)}\right) - \beta\phi\left(\frac{\pi(b \mid x)}{\pi_{\text{ref}}(b \mid x)}\right)\right| \leqslant V_{\max}.$$

Finally, we will measure the theoretical performance using the same type of *single-policy concentrability* as in $\chi$PO.

**Definition 3.4** ($L_1$-Concentrability). The single-policy $L_1$-concentrability coefficient for a policy $\pi$ is given by

$$\mathcal{C}^\pi := \mathbb{E}_\pi\left[\frac{\pi(a|x)}{\pi_{\text{ref}}(a|x)}\right],$$

where we recall that $\mathbb{E}_\pi[\cdot] := \mathbb{E}_{x \sim \rho, a \sim \pi(\cdot|x)}[\cdot]$.

By a direct calculation, one can see $\mathcal{C}^\pi = 2D_{\chi^2}(\pi \| \pi_{\text{ref}}) + 1$, which is extremely useful in the analysis of both $\chi$PO and our next main result on Algorithm 1.

**Theorem 3.5.** *For any given comparator policy $\pi^\star$, there exists a proper choice of $\beta > 0$ such that when Assumptions 3.2 and 3.3 hold, with probability at least $1 - \zeta$, the output of Algorithm 1 satisfies the following suboptimality gaps under* CTL *and* LTC:

$$\text{SG}_{\text{CTL}}(\widehat{\pi}; \pi^\star) \lesssim \kappa(\pi^\star)\left(c(\varepsilon)\sqrt{\frac{\log(|\Pi|/\zeta)}{n}} + \sqrt{\alpha}\right),$$

$$\text{SG}_{\text{LTC}}(\widehat{\pi}; \pi^\star) \lesssim \kappa(\pi^\star)\left(c(\varepsilon)\sqrt{\frac{\log(|\Pi|/\zeta)}{n}} + \sqrt{\alpha \cdot c(\varepsilon)}\right),$$

where $a \lesssim b$ as shorthand for $a = \mathcal{O}(b)$, $c(\varepsilon) = \frac{e^\varepsilon + 1}{e^\varepsilon - 1}$ and $\kappa(\pi^\star) := e^{2R_{\max}} \cdot \frac{V_{\max}}{R_{\max}} \sqrt{\mathcal{C}^{\pi^\star}}$ *is the single-policy concentrability related term.*

*Remark* 3.6. Thanks to the use of RR in CTL and LTC, our algorithm is $\varepsilon$-LDP. Setting $\varepsilon = \infty$ and $\alpha = 0$ in the above utility bounds, leads to the same bound as in $\chi$PO. Moreover, as a by-product, the above theorem also directly gives results for privacy-only and corruption-only settings. Furthermore, it can be easily leveraged to establish bounds for the setting where corruption happens both before and after local privacy with a simple summation of the two bounds above. We stress that, as in Huang et al. (2024), we consider a finite policy class $\Pi$ for the ease of presentation. The extension to an infinite function class can be easily achieved via the standard covering number argument. For example, for a linear reward model in $\mathbb{R}^d$ (or equivalently, a log-linear policy class), $\log |\Pi|$ will roughly be $\widetilde{\mathcal{O}}(d)$.

With the above theorem, several important observations and remarks are in order.

**Interplay between local privacy and corruption.** One can see that under CTL, the impact of local privacy parameter $\varepsilon$ (i.e., the first term) and corruption parameter $\alpha$ (i.e., the second term) is *separable* (additive), while there exists a multiplicative term in LTC, which leads to an additional $\sqrt{c(\varepsilon)} \geqslant 1$ factor. While these are only upper bound results, we tend to believe that the different interplay between local privacy and corruption (i.e., additive vs. multiplicative) indeed exists, especially given the recent similar tight result in mean estimation (Zhou & Zhang, 2024).

**Comparison with prior private alignment.** To the best of our knowledge, Chowdhury et al. (2024b) is the only related work that studies label privacy protection in offline alignment. However, it considers the standard RL-based approach where a reward model is explicitly learned before the policy optimization, rather than our RL-free direct optimization method. More importantly, it only considers the linear reward setting while ours is the first one that establishes formal guarantees for the general function approximation settings with the same (optimal) privacy cost of $c(\varepsilon)$ and a similar single-policy concentrability dependence. Finally, we refer readers to Section 6 for comparisons with one concurrent work (Zhou et al., 2025) on private alignment.

**Comparison with prior robust alignment.** To the best of our knowledge, only Chowdhury et al. (2024a) provides a formal theoretical bound on the suboptimality gap of a robust variant of DPO under a particular type of label corruption. Specifically, it considers the so-called *random-flipping* corruption (i.e., with some *known* probability, the true label is flipped). An astute reader may already observe that this corruption model is weaker than our Huber corruption, and moreover, it is essentially equivalent to label privacy

---

**Algorithm 2** Square$\chi$PO for cDP

1: **Input:** Possibly label corrupted preference dataset $\bar{\mathcal{D}}_{\mathsf{pref}} = \{(x_i, a_i^0, a_i^1, \bar{y}_i)\}_{i=1}^n$, privacy parameter $\varepsilon > 0$, regularization coefficient $\beta > 0$, reference policy $\pi_{\mathsf{ref}}$, $h_{\chi\mathsf{PO},i}$ in (5)

2: Define
$$L(\pi; \bar{\mathcal{D}}_{\mathsf{pref}}) := \sum_{i \in [n]} \left[ 2\sigma \left( \mathsf{clip}_{2R_{\max}} \left[ \beta h_{\chi\mathsf{PO},i} \right] \right) - 1 - \bar{y}_i' \right]^2,$$
where $\bar{y}_i' = 2\bar{y}_i - 1 \in \{1, -1\}$

3: Sample a policy $\widehat{\pi}$ from $\Pi$ via the following distribution
$$P(\pi) \propto \exp \left( -\frac{\varepsilon}{8} \cdot L(\pi; \bar{\mathcal{D}}_{\mathsf{pref}}) \right)$$

4: **Output:** $\widehat{\pi}$

---

noise under RR after a simple reparameterization. Thus, it is in fact more fair to compare it with Theorem 3.5 under $\alpha = 0$. In this context, our main result has two significant improvements over Chowdhury et al. (2024a): (i) Even under the linear model, Chowdhury et al. (2024a) only archives a $\mathcal{O}(1/n^{1/4})$ rate with worse *all-policy concentrability* dependence while ours is the optimal $\mathcal{O}(1/n^{1/2})$ rate with *single-policy concentrability*; (ii) For the general function approximation setting, Chowdhury et al. (2024a) fails to achieve a vanish suboptimality gap as $n \to \infty$ while ours maintains the optimal $\mathcal{O}(1/n^{1/2})$ rate. Another related work is Mandal et al. (2024), which only considers RL-based alignment with linear function approximations under adversary corruption of both prompt (responses) and labels. In contrast, our main focus is RL-free alignment for general function approximations while under label-corruption only. Finally, we refer readers to Section 6 for comparisons with one concurrent work (Zhou et al., 2025) on robust alignment.

### 3.2. Central Model

We now turn to privacy protection in the central model where both the prompt (responses) and labels are sensitive information (cf. cDP in Definition 2.6).

Our proposed algorithm is presented in Algorithm 2, which essentially applies the *exponential mechanism* (McSherry & Talwar, 2007) with our square loss as the score function. The boundedness of our square loss (in contrast to the unboundedness of log-loss) plays a key role in balancing privacy and utility thanks to its bounded *sensitivity*, i.e., changing any single sample at most modify $L(\pi; \bar{\mathcal{D}}_{\mathsf{pref}})$ by 4, which leads to our sampling distribution in Algorithm 2.

We now proceed to present the privacy and utility guarantees of Algorithm 2.

**Theorem 3.7.** *Let $\varepsilon > 0$, Algorithm 2 satisfies $\varepsilon$-DP. For any given comparator policy $\pi^\star$, there exists a proper choice of $\beta > 0$ such that when Assumptions 3.2 and 3.3 hold, with probability at least $1 - \zeta$, the output of Algorithm 2 satisfies the following suboptimality gap under* cDP

$$\mathsf{SG}_{\mathsf{cDP}}(\widehat{\pi}; \pi^\star) \lesssim \kappa(\pi^\star)\left(\left(1 + \frac{1}{\sqrt{\varepsilon}}\right)\sqrt{\frac{\log(|\Pi|/\zeta)}{n}} + \sqrt{\alpha}\right),$$

*where $\kappa(\pi^\star) = e^{2R_{\max}} \cdot \frac{V_{\max}}{R_{\max}}\sqrt{\mathcal{C}^{\pi^\star}}$ is the single-policy concentrability related term.*

With this theorem in hand, several interesting and important observations are in order.

**Interplay between central DP and corruption.** One can first observe that as in CTL, the cost of privacy and corruption is separable (i.e., additive). However, the privacy cost is smaller in the central model than that under the local model.

**Comparison with prior alignment under central DP.** To the best of our knowledge, there are two concurrent works (Chowdhury et al., 2024b; Korkmaz & Brown-Cohen, 2024) that studied RL-based alignment under central DP constraint in the context of a linear reward model in $\mathbb{R}^d$. In particular, they both consider a *weaker* approximate DP constraint (i.e., $\delta > 0$) and establish a privacy cost of $\mathcal{O}\left(\frac{(d\log(1/\delta))^{1/4}}{\sqrt{n\varepsilon}}\right)$. In contrast, our result can handle *general function approximations* with a stronger pure DP guarantee. In fact, if one simply generalizes their approaches by following the non-private counterpart in Zhu et al. (2023) to tackle non-linear functions, it will lead to a strictly suboptimal non-vanishing suboptimality gap. Further, our result under a linear model reduces to a privacy cost of $\mathcal{O}\left(\frac{\sqrt{d}}{\sqrt{n\varepsilon}}\right)$, which has a worse dependence on $d$ (due to the stronger pure DP) while getting rid of the additional $\log(1/\delta)$ factor (which is typically at least on the order of $\log n$).

*Remark 3.8.* It should be clear that Algorithm 2 is not a computationally efficient due to the sampling operation, especially for an infinite class $\Pi$. Hence, we view it as an information-theoretic result, which serves as an important theoretical benchmark for our next step in developing a computationally efficient algorithm. This is indeed a typical path in the private machine learning literature.

## 4. General Preference Model

In this section, we turn to the general preference model, which does not assume preference transitivity as in previous BT-preference model. We will demonstrate that our Square$\chi$PO can be easily extended to this setting based on the self-play framework (Swamy et al., 2024; Gao et al., 2024; Rosset et al., 2024). As already shown in $\chi$PO (Huang et al., 2024) for the standard non-private non-corrupted set-

ting, it is impossible to achieve a single-policy concentrability dependence in the sample complexity bound under the general preference model. Thus, we will aim to achieve a coverage dependence the same as in $\chi$PO under the general preference model, which is somewhat in between single-policy and all-policy concentrability.

In the interest of space, our proposed algorithm for the general preference model under privacy and corruption is presented in Algorithm 3 in appendix. It mainly consists of two key steps: (i) preference model estimation and (ii) policy optimization with self-play. Our modification compared to iterative $\chi$PO in Huang et al. (2024) only lies in the first step, since the labeled data set (which is our corruption and privacy protection target) is only used during the first step while the second step works with an unlabeled dataset $\mathcal{D}_x$.

**(i) Preference model estimation.** Depending on the local or central privacy model, we have two different ways of finding $\widehat{\ell}$. For the local model, $\widehat{\ell}$ is found via a modified least-square regression where an additional factor of $c(\varepsilon)$ is applied in (10), which will essentially reduce to the same loss as in Huang et al. (2024) when $\varepsilon = \infty$ (i.e., no privacy protection). We can now also observe that the loss function under the BT-preference model in Algorithm 1 is simply a specific instantiation of (10) by plugging BT-preference probability (via sigmoid function) into $\ell(x_i, a_i^0, a_i^1)$. Similarly, for the central model, we again use the exponential mechanism to find $\widehat{\ell}$, based on the square loss, which is also a generalization of the loss used in Algorithm 2 under the BT-preference model.

**(ii) Policy optimization with self-play.** With the estimated preference model $\widehat{\ell}$ in hand, we proceed to run policy optimization over a *unlabeled* dataset via self-play, which means that $\widehat{r}^t$ is constructed using the current policy $\pi^t$ (i.e., $b_t \sim \pi^t(x)$). With this $\widehat{r}^t$, our algorithm (which is the same as in Huang et al. (2024)) updates its policy by mirror descent (Nemirovskij & Yudin, 1983) with a mixed regularizer (i.e., $\chi^2$-regularizer and KL-regularizer) over *both* the current policy $\pi^t$ and $\pi_{\mathsf{ref}}$. This type of mirror descent can be rewritten using the same $\chi$PO reparametrization as a regression over a reward difference, leading to the loss $\mathcal{L}_t(\pi; \mathcal{D}_x)$ in (11) with the reparametrization function $f_{\pi,\pi'}^{\beta,\eta}(x, a, b)$ given by

$$\left(1 + \frac{1}{\eta}\right)\beta \cdot h_{\chi\mathsf{PO}\pi}(x, a, b) - \frac{\beta}{\eta} \cdot h_{\chi\mathsf{PO}\pi'}(x, a, b), \quad (8)$$

where $h_{\chi\mathsf{PO}\pi}(x, a, b) := \phi\left(\frac{\pi(a|x)}{\pi_{\mathsf{ref}}(a|x)}\right) - \phi\left(\frac{\pi(b|x)}{\pi_{\mathsf{ref}}(b|x)}\right)$ is essentially the same reparametrization used in the last section (cf. (5)) with $\phi(u) = u + \log u$ being the same as before. At a high level, this policy optimization step can be viewed as a combination of the techniques developed in Gao et al. (2024) (i.e., regression over the reward difference with a

reparametrization trick) and in Chang et al. (2024) (i.e., regularized over both $\pi^t$ and $\pi_{\mathsf{ref}}$). We will provide more intuition on this step in the next section.

### 4.1. Theoretical Guarantees

In this section, we present our main theoretical result on Iterative Square$\chi$PO in Algorithm 3. First, we state the *same* set of assumptions as in Huang et al. (2024).

**Assumption 4.1** (Preference function realizability). The model class $\mathcal{L}$ satisfies $\ell^\star \in \mathcal{L}$ where $\ell^\star$ is the ground truth preference function.

The next assumption is about the policy realizability during each policy update step, which is analogous to Assumption 3.2 in the BT-preference model.

**Assumption 4.2** (Policy realizability for general preferences). For any policy $\pi \in \Pi$ and $\ell \in \mathcal{L}$, the policy class $\Pi$ contains the minimizer of the following regularized optimization objective: $\forall x \in \mathcal{X}$

$$\bar{\pi}(x;\ell,\pi):=\operatorname*{argmax}_{p\in\Delta(\mathcal{X})}\big\{\mathbb{E}_{a\sim p,b\sim\pi(x)}[\ell(x,a,b)]-\mathcal{R}_x(p,\pi_{\mathsf{ref}},\pi)\big\},$$

where the regularizer $\mathcal{R}_x(p, \pi_{\mathsf{ref}}, \pi)$ is given by

$$\mathcal{R}_x(p, \pi_{\mathsf{ref}}, \pi) := \beta D_{f_{\chi_{\mathrm{mix}}}}(p\|\pi_{\mathsf{ref}}(x)) + \frac{\beta}{\eta} B_x(p,\pi),$$

with $D_{f_{\chi_{\mathrm{mix}}}}(p\|q) := D_{\chi^2}(p\|q)+D_{\mathsf{KL}}(p\|q)$ and $B_x(p,q)$ being the Bregman divergence induced by the convex function $F(u) := D_{f_{\chi_{\mathrm{mix}}}}(u\|\pi_{\mathsf{ref}})$, i.e.,

$$B_x(p,q) := F(p) - F(q) - \langle\nabla F(q), p - q\rangle.$$

While it may seem to be complicated, we now pause briefly to provide further intuition on the above optimization by comparing it with $J_\beta^{\chi_{\mathrm{mix}}}$ in Assumption 3.2. We first note that the $\pi$ in $\bar{\pi}(x;\ell,\pi)$ will be $\pi^t$ in our algorithm. Thus, compared with $J_\beta^{\chi_{\mathrm{mix}}}$, the above optimization basically adds another regularization over $\pi^t$ via $B_x(p,\pi^t)$, which directly gives us the reparametrization function in (8)[2] with $\pi' = \pi^t$.

Finally, analogous to Assumption 3.3, we assume that the implicit reward is bounded.

**Assumption 4.3** (Bounded implicit reward difference for general preferences). For a parameter $V_{\max} \geqslant 2$, it holds that for all $\pi, \pi' \in \Pi$, $x \in \mathcal{X}$, and $a, b \in \mathcal{A}$,

$$|f_{\pi,\pi'}^{\beta,\eta}(x, a, b)| \leqslant V_{\max}.$$

Our main guarantee for Algorithm 3 is as follows.

---

[2]Note that the last term in $B_x(p,q)$ will not contribute, since the gradient of it is independent of $p$.

**Theorem 4.4.** *Let $\varepsilon > 0$, Algorithm 3 satisfies $\varepsilon$-LDP or $\varepsilon$-DP, respectively. Let $\mathsf{subopt}(\widehat{\pi}, C) := \max_{\pi \in \Pi} \ell^*(\pi, \widehat{\pi}) - \max_{\pi \in \Pi_C} \ell^*(\pi, \widehat{\pi})$ and $\Pi_C := \{\pi : \max_{x \in \mathcal{X}} D_{\chi^2}(\pi(x) \| \pi_{\mathsf{ref}}(x)) \leqslant C\}$. Then, for any $\zeta \in (0, 1]$ and each setting of CTL, LTC and cDP, under Assumptions 4.1, 4.2 and 4.3, there exists corresponding proper choices of $T, \beta, \eta$ such that with probability $1 - \zeta$, the following bounds hold:*

$$\mathsf{DG}(\widehat{\pi}) \lesssim \min_{C \geqslant 1}\{\mathsf{subopt}(\widehat{\pi}, C) + C \cdot \mathcal{B}\},$$

*where $\mathcal{B} \in \{\mathcal{B}_{\mathsf{CTL}}, \mathcal{B}_{\mathsf{LTC}}, \mathcal{B}_{\mathsf{cDP}}\}$ are defined as*

$$\mathcal{B}_{\mathsf{CTL}} := \left(\mathcal{V}_m + c(\varepsilon)\sqrt{\frac{\log(|\mathcal{L}||\Pi|/\delta)}{n}} + \sqrt{\alpha\log\frac{|\Pi|}{\delta}}\right),$$

$$\mathcal{B}_{\mathsf{LTC}} := \left(\mathcal{V}_m + c(\varepsilon)\sqrt{\frac{\log(|\mathcal{L}||\Pi|/\delta)}{n}} + \sqrt{\alpha c(\varepsilon)\log\frac{|\Pi|}{\delta}}\right),$$

$$\mathcal{B}_{\mathsf{cDP}} := \left(\mathcal{V}_m + \left(1 + \frac{1}{\sqrt{\varepsilon}}\right)\sqrt{\frac{\log(|\mathcal{L}||\Pi|/\delta)}{n}} + \sqrt{\alpha\log\frac{|\Pi|}{\delta}}\right),$$

*where $\mathcal{V}_m := V_{\max}\frac{\log(|\Pi|/\delta)}{\sqrt{m}}$.*

*Remark 4.5.* We remark again that this is the first set of results for private and robust alignment under a general preference model.

## 5. Key Techniques Highlight

In this section, we would like to highlight a key common technique behind all the results in previous sections. In particular, all of our sample complexity bounds build upon the following lemma that characterize generazation error bounds of least-square regression under CTL, LTC or cDP.

**Lemma 5.1** (Informal statement of Lemma B.1). *Let $\{(u_i, y_i')\}_{i=1}^n$ be a clean dataset and $\mathcal{H}$ be a hypothesis class such that realizability holds ($h^* \in \mathcal{H}$). Define generalization error for any $\widehat{h}$ as*

$$\mathsf{err}_{\mathsf{gen}}^2 := \mathbb{E}_{u \sim \rho'}[(\widehat{h}(u) - h^*(u))^2],$$

*for feature distribution $\rho'$. Then, with probability at least $1 - \zeta$, we have*

*1. Under CTL and LTC, given $\{(u_i, z_i')\}_{i=1}^n$ as input dataset, $\widehat{h} = \operatorname{argmin}_{h \in \mathcal{H}} \sum_{i=1}^n (h(u_i) - c(\varepsilon)z_i')^2$ achieves*

$$\mathsf{err}_{\mathsf{gen,CTL}}^2 \lesssim c(\varepsilon)^2 \cdot \frac{\log(|\mathcal{H}|/\zeta)}{n} + \alpha,$$

$$\mathsf{err}_{\mathsf{gen,LTC}}^2 \lesssim c(\varepsilon)^2 \cdot \frac{\log(|\mathcal{H}|/\zeta)}{n} + \alpha \cdot c(\varepsilon).$$

*2. Under cDP, given $\{(u_i, \bar{y}_i')\}_{i=1}^n$ as input dataset, running exponential mechanism using square loss over $\bar{y}_i'$ yields*

$$\mathsf{err}_{\mathsf{gen,cDP}}^2 \lesssim \frac{\log(|\mathcal{H}|/\zeta)}{n} + \frac{\log(|\mathcal{H}|/\zeta)}{n\varepsilon} + \alpha.$$

The above result is a nontrivial extension of the standard findings in (Song et al., 2022) to the private and corrupted settings. Given the widespread use of least-squares regression oracles in offline, online, and hybrid RL (Agarwal et al., 2019), we believe this result can be readily applied to drive new advancements in the private and corrupted scenarios.

## 6. Discussion

In this section, we first provide a detailed discussion on the concurrent work (Zhou et al., 2025) on private and robust offline alignment, which shares similar motivations but has the following key differences. First, Zhou et al. (2025) only focuses on the linear model with BT-preference, while we consider general function approximations for BT-preference as well as a general preference model. Second, Zhou et al. (2025) only considers local DP, while we also consider central DP. Third, Zhou et al. (2025) considers a strong corruption model while we consider a slightly weaker model, i.e., Huber corruption model. This gives a different term regarding the interplay between privacy and corruption, i.e., $c(\varepsilon)\sqrt{\alpha}$ vs. $\sqrt{c(\varepsilon)\alpha}$. We also believe that the dependence on $\alpha$ in Lemma 5.1 can be improved to $\alpha^2$ by leveraging the Huber corruption property[3]. Second, although we mainly focus on the theory in the main body, we have also managed to conduct some experiments as proof-of-concept, see Appendix E for details.

## 7. Conclusion

We introduced Square$\chi$PO, a novel offline alignment method that achieves state-of-the-art theoretical guarantees in the presence of noisy labels caused by privacy protections and/or adversarial corruption. Our algorithm can handle both BT-preference and general preference models. While our primary focus is theoretical, Square$\chi$PO remains practical and easy to implement, requiring only a minor modification to $\chi$PO and DPO. Future work will focus on comprehensive empirical evaluations to further validate our findings.

## Acknowledgements

XZ is supported in part by NSF CNS-2153220 and CNS-2312835.

## Impact Statement

This paper presents work whose goal is to advance the field of Machine Learning. There are many potential societal consequences of our work, none which we feel must be specifically highlighted here.

---

[3]In fact, we are working on a new paper that will have a more thorough discussion. Stay tuned.

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

# A. Additional Related Work

The alignment problem has been extensively studied in the previous literature (Yu et al., 2021; Ziegler et al., 2019; Stiennon et al., 2020; Bai et al., 2022a; Shin et al., 2023; Zhan et al., 2023; Mandal et al., 2024). Besides the private or robust alignment related work we mentioned in the main text, we refer the readers to Sun et al. (2024a) for more general trustworthiness in large language models and to Xiao & Zhu (2025); Touvron et al. (2023) for comprehensive surveys on large language models. Here, we discuss some additional related work.

**Alignment with Human Feedback.** The most fundamental method to align LLM is Reinforcement Learning from Human Feedback (RLHF) (Christiano et al., 2017; Ouyang et al., 2022), which has been practically used in OpenAI (2022); Sun et al. (2024b); Bai et al. (2022a;b). Instead of fine-tuning models by training a reward model from human feedback and optimizing policy using Reinforcement Learning (e.g., Proximal policy optimization (PPO) (Schulman et al., 2017)), Direct Preference Optimization (DPO) (Rafailov et al., 2023) simplifies alignment by directly optimizing the policy using human preference data. This approach bypasses the need for a reward model and reinforcement learning method, resulting in a more stable and efficient training process (Abdin et al., 2024). In the following, we divide related work on alignment with human feedback based on different perspectives:

- **Extended works from DPO.** Taking DPO as a starting point, many preference optimization variants have emerged to improve efficiency, stability, adaptability, or other properties. Relevant examples are Chi-Squared Preference Optimization ($\chi$PO) (Huang et al., 2024), Rejection Sampling Optimization (RSO) (Liu et al., 2023), Identity Preference Optimization (IPO) (Azar et al., 2024), $\Psi$PO (Azar et al., 2024), generalized preference optimization (GPO) (Tang et al., 2024), Direct Nash Optimization (DNO) (Rosset et al., 2024), Self-Play Preference Optimization (SPPO) (Wu et al., 2024a), and Exploratory Preference Optimization (XPO) (Xie et al., 2024). Our Square$\chi$PO is a variant of $\chi$PO, where the main difference is in the loss function—more on this in the next bullet point.

- **The role of loss function.** Our Square$\chi$PO is mainly different from the original $\chi$PO in the loss function used to estimate the policy, changed from log-loss to least square loss over probabilities. Compared to the log-loss, the square loss provides a more interpretable measure of error, avoids extreme gradient values for small probability estimates, and ensures numerical stability. Wang et al. (2024a) explores how different loss functions affect the sample efficiency and adaptivity in classification and RL problems. We remark that the use of the square loss is not by any means new in RL. For example, we have temporal-difference (TD) learning with squared loss for regression (Jin et al., 2021a; Xie et al., 2022) and Fitted Q-Iteration (FQI) (Munos & Szepesvári, 2008; Chen & Jiang, 2019), which uses least-squares to approximate the Bellman backup. Thus, we believe that our new generalization error bound can be useful when one aims to extend those problems to private and robust scenarios.

- **Type of regularization divergence.** The objective function of preference optimization can be generally written as *(reward) loss* + (regularization) *penalty* (Xiao & Zhu, 2025). A number of different regularizers have been proposed in the literature. Wang et al. (2023a) proposes a generalized approach, $f$-DPO, by using $f$-divergences for the regularization term, to integrate a variety of popular divergences. Our mixed $\chi^2$ divergence in Square$\chi$PO can be viewed as a special case of $f$-DPO, and it can provably alleviate overoptimization and achieve sample-complexity guarantees based on single-policy concentrability (Huang et al., 2024). Notably, $\chi^2$-regularization has been used in a number of RL works to derive single-policy concentrability guarantee (Wang et al., 2024b; Gabbianelli et al., 2024; Duan et al., 2020; Zhan et al., 2022; Amortila et al., 2024b; Zhu & Zhang, 2024; Lee et al., 2021; Ma et al., 2022a;b). Xiao et al. (2024) introduces a new regularizer called preference matching divergence which helps the LLM balance response diversification and reward maximization. Moreover, Liu et al. (2024) shows that the SFT Loss is implicitly an adversarial regularizer in RLHF, that provably mitigates overoptimization.

- **Coverage coefficients (or concentrability coefficients).** Coverage, a concept that captures how the training data "covers" the test distribution, has played a fundamental role in offline RL (Munos & Szepesvári, 2008; Xie et al., 2021a; Uehara & Sun, 2021; Zhan et al., 2022), offline-online (hybrid) RL (Ross & Bagnell, 2012; Xie et al., 2021b; Song et al., 2022; Amortila et al., 2024a; Song et al., 2024), and online RL (Kakade & Langford, 2002; Bagnell et al., 2003; Xie et al., 2022). The sub-optimality guarantees of Square$\chi$PO obtained under the BT-preference model are based on the *single-policy concentrability*, that is, the data only needs to have a good cover over the chosen comparator policy. This is the gold standard in offline reinforcement learning due to being more effective compared with *all-policy concentrability*, which requires the offline data distribution to provide good coverage over the state distributions induced by *all* candidate policies.

**Privacy and robustness interplay.** The interaction of privacy and robustness has been investigated in many machine learning tasks. In the multi-arm bandits problem, the interaction of central DP and Huber corruption on rewards is investigated in Wu et al. (2024b), while the different orders of LDP and Huber corruption of rewards feedback of bandits have been studied in Zhou & Zhang (2024). Charisopoulos et al. (2023) study the problem of linear bandits problem, where the rewards are under LDP and Huber model. In statistical learning, there are many works that studied the interaction of privacy and robustness in different tasks (e.g., Kamath, 2024; Li et al., 2023; Chhor & Sentenac, 2023). Other works have studied the possibility of privacy might imply robustness or vice-versa. For example, Georgiev & Hopkins (2022) concludes that private mechanisms are automatically robust in many statistics problems. In contrast, Hopkins et al. (2023) shows adversarial robustness implies differential privacy in statistical estimation. In this paper, we investigate both central DP and local DP interacting with Huber contamination model in the offline alignment problem.

**Private online RL.** In contrast to the offline RL setting in this paper, there has been a recent line of work on private (and robust) online RL under various settings and DP models, including MABs (e.g., Mishra & Thakurta (2015); Sajed & Sheffet (2019); Chowdhury & Zhou (2022b); Wu et al. (2023); Ren et al. (2020)), structured (contextual) bandits (e.g., Shariff & Sheffet (2018); Zheng et al. (2020); Chowdhury & Zhou (2022c); Li et al. (2022); Zhou & Tan (2021)) and RL (e.g., Vietri et al. (2020); Garcelon et al. (2021); Chowdhury & Zhou (2022a); Qiao & Wang (2023); Zhou (2022)). One main limitation of these works is that they only consider tabular, linear (or kernerlized) function approximations, while general function approximation result is still missing.

# B. Generalization Bounds of Least-Square Regression under Privacy and Corruption

In this section, we provide a detailed version of our main techniques, i.e., generalization error bound of least-square regression under privacy constraints and corruption. We mainly focus on the case where the response variable is binary, given its immediate application in our scenarios. However, it can be easily generalized to the continuous case via random rounding, see Zhou & Zhang (2024).

**Lemma B.1.** *Let $\{(u_i, y_i')\}_{i=1}^n$ be a clean dataset of $n$ points where each point is independently sampled from $u_i \sim \rho'$ and $y_i' \sim p(\cdot|u_i) := h^*(u_i) + \eta_i$, where $\{\eta_i\}_{i=1}^n$ are independent random variables such that $\mathbb{E}[y_i'|u_i] = h^*(u_i)$ and $y_i' \in \{-1, 1\}$. Let $\mathcal{H} : \mathcal{U} \to [-1, 1]$ be a class of real valued functions such that $h^* \in \mathcal{H}$, i.e., we assume realizability. Define the generalization error bounds for a learning algorithm's output $\widehat{h}$ as*

$$\mathsf{err}_{\mathsf{gen}}^2 := \mathbb{E}_{u \sim \rho'}[(\widehat{h}(u) - h^*(u))^2].$$

*Then, we have the following results across different settings:*

1. *Under* CTL *or* LTC *where the observed dataset is $\{(u_i, z_i')\}_{i=1}^n$ (with $z_i' \in \{-1, 1\}$) that is generated according to* CTL *or* LTC *(Definition 2.3), the least-square regression solution $\widehat{h} = \operatorname{argmin}_{h \in \mathcal{H}} \sum_{i=1}^n (h(u_i) - c(\varepsilon)z_i')^2$ (with $c(\varepsilon) = \frac{e^\varepsilon + 1}{e^\varepsilon - 1}$) satisfies with probability at least $1 - \zeta$*

$$\mathsf{err}_{\mathsf{gen,CTL}}^2 \lesssim c(\varepsilon)^2 \cdot \frac{\log(|\mathcal{H}|/\zeta)}{n} + \alpha,$$

$$\mathsf{err}_{\mathsf{gen,LTC}}^2 \lesssim c(\varepsilon)^2 \cdot \frac{\log(|\mathcal{H}|/\zeta)}{n} + \alpha \cdot c(\varepsilon).$$

2. *Under* cDP *where the observed dataset is $\{(u_i, \bar{y}_i')\}_{i=1}^n$ (with $\bar{y}_i' \in \{-1, +1\}$) that is generated according* cDP *(Definition 2.6), sampling $\widehat{h}$ via the following exponential mechanism:*

$$P(h) \propto \exp\left(-\frac{\varepsilon}{8} \cdot L(h)\right) \ \forall h \in \mathcal{H},$$

*with $L(h) := \sum_{i \in [n]} [h(u_i) - \bar{y}_i']^2$, yields that*

$$\mathsf{err}_{\mathsf{gen,cDP}}^2 \lesssim \frac{\log(|\mathcal{H}|/\zeta)}{n} + \frac{\log(|\mathcal{H}|/\zeta)}{n\varepsilon} + \alpha.$$

*Remark* B.2. This result can be viewed as a nontrivial generalization of the standard one in Song et al. (2022) to the private and corrupted scenarios.

A key lemma in our proof is the following form of Freedman's inequality.

**Lemma B.3** (Theorem 1 in Beygelzimer et al. (2011)). *Let $\{u_i\}_{i \leq n}$ be a real-valued martingale difference sequence adapted to a filtration $\{\mathcal{F}_i\}_{i \leq n}$. If $u_i \leq R$ almost surely, then for any $\eta \in (0, 1/R]$, with probability at least $1 - \zeta$,*

$$\sum_{i=1}^{n} u_i \leq \eta(e-2) \sum_{i=1}^{n} \mathbb{E}_{i-1}[u_i^2] + \frac{\log(1/\zeta)}{\eta},$$

*where $\mathbb{E}_{i-1}[\cdot] := \mathbb{E}[\cdot | \mathcal{F}_{i-1}]$.*

We actually do not need the martingale structure, but for simplicity we will still use the above well-known lemma.

Now we are ready to prove our generalization bound.

*Proof of Lemma B.1.* We start with CTL and the other two are similar. For any fixed $h \in \mathcal{H}$, we define

$$U_i^h := (h(u_i) - c(\varepsilon)z_i')^2 - (h^*(u_i) - c(\varepsilon)z_i')^2.$$

Also, define

$$D_i^h := \mathbb{E}[U_i^h] - U_i^h.$$

Given that the $D_i^h$ are i.i.d. (due to Huber corruption) and with mean equal to zero, they are also a martingale difference sequence. Moreover, the $U_i^h$ are also i.i.d., hence any application of $\mathbb{E}_{i-1}[\cdot]$ to any point-wise function of these random variables will be equal to $\mathbb{E}[\cdot]$ on the same function.

We further notice that

$$\mathbb{E}[(D_i^h)^2] \leq \mathbb{E}[(U_i^h)^2] = \mathbb{E}[(h(u_i) - h^*(u_i))^2 (h(u_i) + h^*(u_i) - 2c(\varepsilon)z_i')^2]$$
$$\lesssim c(\varepsilon)^2 \cdot \mathbb{E}[(h(u_i) - h^*(u_i))^2],$$

where the last step holds by the boundedness of $z_i'$ and $h \in \mathcal{H}$. Moreover, let $\bar{y}_i$ be the intermediate corrupted label, we have

$$\mathbb{E}[U_i^h] = \mathbb{E}[(h(u_i) - h^*(u_i))(h(u_i) + h^*(u_i) - 2c(\varepsilon)z_i')]$$
$$= \mathbb{E}[(h(u_i) - h^*(u_i))(h(u_i) + h^*(u_i) - 2c(\varepsilon)z_i' + 2\bar{y}_i - 2\bar{y}_i + 2y_i' - 2y_i')]$$
$$= \underbrace{\mathbb{E}[(h(u_i) - h^*(u_i))(-2c(\varepsilon)z_i' + 2\bar{y}_i)]}_{\mathcal{T}_{\text{privacy}}} + \underbrace{\mathbb{E}[(h(u_i) - h^*(u_i))(2y_i' - 2\bar{y}_i)]}_{\mathcal{T}_{\text{corruption}}}$$
$$+ \underbrace{\mathbb{E}[(h(u_i) - h^*(u_i))(h(u_i) + h^*(u_i) - 2y_i')]}_{\mathcal{T}_{\text{standard}}}.$$

We are going to bound each of them. For $\mathcal{T}_{\text{privacy}}$, due to the generation process of $z_i'$ via random response over $\bar{y}_i$ and the fact that each privacy noise in random response is independent of all other randomness, we have $\mathcal{T}_{\text{privacy}} = 0$. For $\mathcal{T}_{\text{standard}}$, due to the fact that $\mathbb{E}[y_i'|u_i] = h^*(u_i)$, we have

$$\mathcal{T}_{\text{standard}} = \mathbb{E}[(h(u_i) - h^*(u_i))^2].$$

Combining all three terms, yields that

$$\mathbb{E}[U_i^h] = \mathbb{E}[(h(u_i) - h^*(u_i))^2] + \mathbb{E}[(h(u_i) - h^*(u_i))(2y_i' - 2\bar{y}_i)].$$

Then, applying Lemma B.3 to $\{D_i^h\}_{i \leq n}$ with a proper choice of $\eta$, we have

$$\sum_i \mathbb{E}[(h(u_i) - h^*(u_i))^2] + \sum_i \mathbb{E}[(h(u_i) - h^*(u_i))(2y_i' - 2\bar{y}_i)]$$
$$\lesssim \sum_i U_i^h + \frac{1}{2} \sum_i \mathbb{E}[(h(u_i) - h^*(u_i))^2] + c(\varepsilon)^2 \cdot \log(1/\zeta).$$

Re-arranging it leads to

$$\sum_i \mathbb{E}[(h(u_i) - h^*(u_i))^2] \lesssim \sum_i U_i^h + c(\varepsilon)^2 \cdot \log(1/\zeta) + \sum_i \mathbb{E}[(h(u_i) - h^*(u_i))(2\bar{y}_i - 2y_i')].$$

Using a union bound over all $h \in \mathcal{H}$, we have that

$$\sum_i \mathbb{E}[(h(u_i) - h^*(u_i))^2] \lesssim \sum_i U_i^h + c(\varepsilon)^2 \cdot \log(|\mathcal{H}|/\zeta) + \sum_i \mathbb{E}[(h(u_i) - h^*(u_i))(2\bar{y}_i - 2y_i')], \ \forall h \in \mathcal{H}.$$

Let's now use this result for $\widehat{h}$, noting that $\sum_i U_i^{\widehat{h}} \leq 0$. So, we have

$$\sum_i \mathbb{E}[(\widehat{h}(u_i) - h^*(u_i))^2] \lesssim c(\varepsilon)^2 \cdot \log(|\mathcal{H}|/\zeta) + \sum_i \mathbb{E}[(\widehat{h}(u_i) - h^*(u_i))(2\bar{y}_i - 2y_i')]$$
$$\lesssim c(\varepsilon)^2 \cdot \log(|\mathcal{H}|/\zeta) + \alpha n,$$

where the last step follows from $\alpha$-Huber corruption. Finally, given the i.i.d corruption, we can divide both sides by $n$, leading to

$$\mathbb{E}_{u \sim \rho}[(\widehat{h}(u) - h^*(u))^2] \lesssim c(\varepsilon)^2 \cdot \frac{\log(|\mathcal{H}|/\zeta)}{n} + \alpha,$$

which completes the proof for CTL.

**LTC case.** It follows the same proof flow as above and we highlight the different steps only. Now, let $\widetilde{y}_i$ be the intermediate privatized label, we have

$$\mathbb{E}[U_i^h] = \mathbb{E}[(h(u_i) - h^*(u_i))(h(u_i) + h^*(u_i) - 2c(\varepsilon)z_i')]$$
$$= \mathbb{E}[(h(u_i) - h^*(u_i))((h(u_i) + h^*(u_i)) - 2c(\varepsilon)(z_i' - \widetilde{y}_i + \widetilde{y}_i))]$$
$$= \underbrace{\mathbb{E}[(h(u_i) - h^*(u_i))(-2c(\varepsilon)(z_i' - \widetilde{y}_i))]}_{\mathcal{T}_{\text{corruption}}} + \underbrace{\mathbb{E}[(h(u_i) - h^*(u_i))(-2c(\varepsilon)\widetilde{y}_i + h(u_i) + h^*(u_i))]}_{\mathcal{T}_{\text{privacy}}}.$$

By the unbiased property of $c(\varepsilon)\widetilde{y}_i$ due to randomized response, we have

$$\mathcal{T}_{\text{privacy}} = \mathbb{E}[(h(u_i) - h^*(u_i))^2].$$

Then, again, applying Lemma B.3 to $\{D_i^h\}_{i \leq n}$ with a proper choice of $\eta$, we have

$$\sum_i \mathbb{E}[(h(u_i) - h^*(u_i))^2] + \sum_i \mathbb{E}[(h(u_i) - h^*(u_i))(-2c(\varepsilon)(z_i' - \widetilde{y}_i))]$$
$$\lesssim \sum_i U_i^h + \frac{1}{2}\sum_i \mathbb{E}[(h(u_i) - h^*(u_i))^2] + c(\varepsilon)^2 \cdot \log(1/\zeta).$$

Re-arranging it leads to

$$\sum_i \mathbb{E}[(h(u_i) - h^*(u_i))^2] \lesssim \sum_i U_i^h + c(\varepsilon)^2 \cdot \log(1/\zeta) + \mathbb{E}[(h(u_i) - h^*(u_i))(2c(\varepsilon)(z_i' - \widetilde{y}_i))],$$

where the last term is the key difference with an additional $c(\varepsilon)$ factor. Following the same argument as in CTL, we have that under LTC

$$\mathbb{E}_{u \sim \rho}[(\widehat{h}(u) - h^*(u))^2] \lesssim c(\varepsilon)^2 \cdot \frac{\log(|\mathcal{H}|/\zeta)}{n} + \alpha c(\varepsilon).$$

**cDP case.** For any fixed $h \in \mathcal{H}$, we define

$$U_i^h := (h(u_i) - \bar{y}_i')^2 - (h^*(u_i) - \bar{y}_i')^2.$$

As in the first case, the $U_i^h$ are i.i.d. Moreover, the random variables

$$D_i^h := \mathbb{E}[U_i^h] - U_i^h.$$

are i.i.d. and have zero mean. We further notice that

$$\mathbb{E}[(D_i^h)^2] \leq \mathbb{E}[(U_i^h)^2] = \mathbb{E}[(h(u_i) - h^*(u_i))^2(h(u_i) + h^*(u_i) - \bar{y}_i')^2]$$
$$\lesssim \mathbb{E}[(h(u_i) - h^*(u_i))^2],$$

where the last step holds by the boundedness of $\bar{y}_i'$ and $h \in \mathcal{H}$. Moreover, let $y_i'$ be the raw uncorrupted label, we have

$$\mathbb{E}[U_i^h] = \mathbb{E}[(h(u_i) - h^*(u_i))(h(u_i) + h^*(u_i) - 2\bar{y}_i')]$$
$$= \mathbb{E}[(h(u_i) - h^*(u_i))(h(u_i) + h^*(u_i) - 2\bar{y}_i' + 2y_i' - 2y_i')]$$
$$= \underbrace{\mathbb{E}[(h(u_i) - h^*(u_i))(2y_i - 2\bar{y}_i')]}_{\mathcal{T}_{\text{corruption}}} + \underbrace{\mathbb{E}[(h(u_i) - h^*(u_i))(h(u_i) + h^*(u_i) - 2y_i')]}_{\mathcal{T}_{\text{standard}}}$$
$$= \underbrace{\mathbb{E}[(h(u_i) - h^*(u_i))(2y_i' - 2\bar{y}_i')]}_{\mathcal{T}_{\text{corruption}}} + \sum_i \mathbb{E}[(h(u_i) - h^*(u_i))^2].$$

Now, applying Lemma B.3 to $\{D_i^h\}_{i \leq n}$ with a proper choice of $\eta$ and re-arranging plus union bound, we have for all $h \in \mathcal{H}$

$$\sum_i \mathbb{E}[(h(u_i) - h^*(u_i))^2] \lesssim \sum_i U_i^h + \log(|\mathcal{H}|/\zeta) + \mathbb{E}[(h(u_i) - h^*(u_i))(2(\bar{y}_i' - y_i'))].$$

Now, compared to CTL and LTC where $\sum_i U_i^{\widehat{h}} \leq 0$, we now have to leverage the utility of the exponential mechanism (McSherry & Talwar, 2007). In particular, let $h' \in \arg \min L(h) = \arg \min \sum_{i \in [n]} [h(u_i) - \bar{y}_i']^2$, then we have with probability at least $1 - \zeta$, for the output of $\widehat{h}$ by the exponential mechanism

$$\sum_{i \in [n]} [\widehat{h}(u_i) - \bar{y}_i]^2 \leq \sum_{i \in [n]} [h'(u_i) - \bar{y}_i']^2 + \frac{\log(|\mathcal{H}|/\zeta)}{\varepsilon},$$

which implies that $\sum_i U_i^{\widehat{h}} \leq \frac{\log(|\mathcal{H}|/\zeta)}{\varepsilon}$.

Finally, following the same argument as before, we arrive at

$$\mathbb{E}_{u \sim \rho}[(\widehat{h}(u) - h^*(u))^2] \lesssim \frac{\log(|\mathcal{H}|/\zeta)}{n} + \frac{\log(|\mathcal{H}|/\zeta)}{n\varepsilon} + \alpha,$$

which completes the proof for the cDP case. $\qquad\square$

## C. Additional Details on Section 3

In this section, we provide the proof of our main results in Section 3, which directly follows from Theorem C.1 and Lemma C.2 below. As we already mentioned, our proof is modular once we have the generalization error bounds. To provide more intuition on this, we first present the following meta theorem, which is a simple adaptation from the proof in Huang et al. (2024) to our Square$\chi$PO.

**Theorem C.1** (Meta Theorem for Square$\chi$PO under BT). *Under the BT-preference model, let Assumptions 3.2 and 3.3 hold. Define $\widehat{r}(x, a) := \beta\phi\left(\frac{\widehat{\pi}(a|x)}{\pi_{\text{ref}}(a|x)}\right)$ for any output policy of Square$\chi$PO (Algorithm 1 or Algorithm 2). Then, we have*

$$J(\pi^\star) - J(\widehat{\pi}) \leq \frac{2V_{\max}}{R_{\max}}\sqrt{C^{\pi^\star} \cdot \text{err}_{\text{stat}}^2} + \beta \cdot C^{\pi^\star} + 2\beta^{-1} \cdot \frac{V_{\max}^2 \text{err}_{\text{stat}}^2}{R_{\max}^2},$$

*where*

$$\text{err}_{\text{stat}}^2 = \mathbb{E}_{\pi_{\text{ref}}, \pi_{\text{ref}}}\left[\left(\text{clip}_{2R_{\max}}[\widehat{\Delta}] - \text{clip}_{2R_{\max}}[\Delta^\star]\right)^2\right],$$

*with* $\widehat{\Delta} := \widehat{r}(x,a) - \widehat{r}(x,b)$ *and* $\Delta^\star := r^\star(x,a) - r^\star(x,b)$. *Furthermore, by taking* $\beta = \sqrt{\frac{2}{\mathcal{C}^{\pi^\star}}} \cdot \frac{V_{\max}\mathsf{err}_{\mathsf{stat}}}{R_{\max}}$, *we have*

$$J(\pi^\star) - J(\widehat{\pi}) \lesssim \frac{V_{\max}}{R_{\max}} \sqrt{\mathcal{C}^{\pi^\star} \cdot \mathsf{err}^2_{\mathsf{stat}}} \ .$$

*Proof.* The above result largely follows from the proof of Theorem E.1 in Huang et al. (2024). The key in their proof is the translation from working with policy to working with the implicit reward $\widehat{r}$ define above, i.e., Lemma E.2 in Huang et al. (2024). With this, one can follow the standard proof for RLHF to arrive at the above result by relying on the fact that $\mathcal{C}^\pi = 2D_{\chi^2}(\pi\|\pi_{\mathsf{ref}}) + 1$. Note that since our Square$\chi$PO uses the same re-parametrization function $\phi$ as in $\chi$PO, so the above argument via their Lemma E.2 still works. One subtlety here is that for cDP, our algorithm for finding $\widehat{\pi}$ is no longer a minimization problem. However, this is still fine since Lemma E.2 holds for any valid policy. $\square$

With this meta theorem, all we need to do is to bound $\mathsf{err}^2_{\mathsf{stat}}$ under CTL, LTC and cDP, respectively, which will directly lead to our main results in Theorem 3.5 and Theorem 3.7. At a high level, without clipping, $\mathsf{err}^2_{\mathsf{stat}}$ can be bounded by directly leveraging our generalization error bound under realizability (Lemma B.1) and mean-value theorem to handle the non-linearity of $\sigma(\cdot)$ function. Here, the main reason for us to do the clipping is to ensure that the cost due to non-linearity is $O(e^{cR_{\max}})$ (for some constant $c > 0$) rather than the worse bound $O(e^{cV_{\max}})$. Due to this additional clipping, we have to carefully show that clipping will not impact our analysis, by showing that *realizability* is still satisfied. This should not be a surprise given the boundedness of $r^*$ and all we need in the analysis is the *reward difference*.

Formally, we have the following bounds on $\mathsf{err}^2_{\mathsf{stat}}$ under CTL, LTC and cDP, respectively.

**Lemma C.2.** *Under the same conditions of Theorem C.1,* $\mathsf{err}^2_{\mathsf{stat}}$ *for* Square$\chi$PO *in Algorithms 1 and 2 satisfies the following bounds:*

$$\mathsf{err}^2_{\mathsf{stat,CTL}} \lesssim e^{4R_{\max}}\left(c(\varepsilon)^2 \cdot \frac{\log(|\Pi|/\zeta)}{n} + \alpha\right),$$

$$\mathsf{err}^2_{\mathsf{stat,LTC}} \lesssim e^{4R_{\max}}\left(c(\varepsilon)^2 \cdot \frac{\log(|\Pi|/\zeta)}{n} + \alpha \cdot c(\varepsilon)\right),$$

$$\mathsf{err}^2_{\mathsf{stat,cDP}} \lesssim e^{4R_{\max}}\left(\frac{\log(|\Pi|/\zeta)}{n} + \frac{\log(|\Pi|/\zeta)}{n\varepsilon} + \alpha\right) \ .$$

*Proof.* **Local model.** By using the implicit reward function, we can re-write Step 3 in Algorithm 1 as

$$\widehat{r} = \operatorname*{argmin}_{r \in \mathcal{R}_\Pi} \sum_{i \in [n]} \left[2\sigma\left(\mathsf{clip}_{2R_{\max}}\left[r(x_i, a_i^1) - r(x_i, a_i^0)\right]\right) - 1 - c(\varepsilon)\bar{z}_i\right]^2,$$

where

$$\mathcal{R}_\Pi := \left\{r(x,a) = \beta \cdot \phi\left(\frac{\pi(a \mid x)}{\pi_{\mathsf{ref}}(a \mid x)}\right) : \pi \in \Pi\right\},$$

and $\bar{z}_i = 2z_i - 1 \in \{1, -1\}$. In order to apply our generalization error bound in Lemma B.1, we can do the following mappings: for any $r \in \mathcal{R}_\Pi$, we map it to a function $h \in \mathcal{H}$ with $|\mathcal{H}| \leq |\Pi|$ via $h(u_i) := 2\sigma\left(\mathsf{clip}_{2R_{\max}}\left[r(x_i, a_i^1) - r(x_i, a_i^0)\right]\right) - 1 \in [-1, 1]$ with $u_i = (x_i, a_i^1, a_i^0)$. Moreover, the label $\bar{z}_i$ is mapped to $z_i'$ and the distribution over prompts and actions is mapped to $\rho'$ in Lemma B.1. With such a mapping, all we need to check is the realizability, i.e., there exists an $h^* \in \mathcal{H}$ defined below such that for the true clean preference label $y_i \in \{0, 1\}$

$$\mathbb{E}[y_i'|u_i] = \mathbb{E}[2y_i - 1|u_i] = h^*(u_i) := 2\sigma\left(\mathsf{clip}_{2R_{\max}}\left[\widetilde{r}^*(x_i, a_i^1) - \widetilde{r}^*(x_i, a_i^0)\right]\right) - 1, \tag{9}$$

where $h^*$ is mapped from $\widetilde{r}^* := \beta \cdot \phi\left(\frac{\pi_\beta^*(a|x)}{\pi_{\mathsf{ref}}(a|x)}\right)$, which satisfies $\widetilde{r}^* \in \mathcal{R}_\Pi$ (hence $h^* \in \mathcal{H}$), because of policy realizability $\pi_\beta^* \in \Pi$. To verify that (9) indeed holds, we note that

$$\mathsf{clip}_{2R_{\max}}\left[\widetilde{r}^\star(x,a) - \widetilde{r}^\star(x,b)\right] = \mathsf{clip}_{2R_{\max}}\left[r^\star(x,a) - r^\star(x,b)\right] = r^\star(x,a) - r^\star(x,b),$$

where the first equality holds by the folklore fact that $\widetilde{r}^*$ is equivalent to $r^*$ up to an action-independent normalization factor, which gets canceled in the reward difference, and the second equality holds by the boundedness of true reward $r^* \in [0, R_{\max}]$.

Applying $\sigma$ function to both sides and noting that under the BT-preference model $\mathbb{E}[y_i|u_i] = \sigma(r^*(x_i, a_i^1) - r^*(x_i, a_i^0))$, yields the realizability condition in (9).

Thus, we can now safely apply Lemma B.1 to obtain results for the local model. In particular, for CTL, we have

$$\mathbb{E}_{u\sim\rho}[(\widehat{h}(u) - h^*(u))^2] = \mathbb{E}_{\pi_{\text{ref}}, \pi_{\text{ref}}}\left[\left(\sigma(\text{clip}_{2R_{\max}}[\widehat{\Delta}]) - \sigma(\text{clip}_{2R_{\max}}[\Delta^\star])\right)^2\right] \lesssim c(\varepsilon)^2 \cdot \frac{\log(|\Pi|/\zeta)}{n} + \alpha,$$

which directly leads to our conclusion by a standard mean-value theorem argument (cf. Lemma C.3 below) to get rid of $\sigma$ function. The same argument applies to LTC case.

**Central model.** The proof for cDP is similar. By using the implicit reward function, we can see that Step 3 in Algorithm 2 is equivalent to running the exponential mechanism with

$$P(r) \propto \exp\left(-\frac{\varepsilon}{8} \cdot L(r)\right) \forall r \in \mathcal{R}_\Pi,$$

with $L(r) := \sum_{i\in[n]}[2\sigma\left(\text{clip}_{2R_{\max}}\left[r(x_i, a_i^1) - r(x_i, a_i^0)\right]\right) - 1 - \bar{y}_i']^2$.

Then, with the same mapping argument as in the local model, we can verify the realizability condition. Hence, we can apply Lemma B.1 along with Lemma C.3 to arrive at the final result. □

**Lemma C.3.** *For $z, z' \in [-R, R]$ and $R \geqslant 1$, by mean-value theorem we have*

$$|z - z'| \leqslant (e^{-R} + 2 + e^R) \cdot |\sigma(z) - \sigma(z')|,$$

*where $\sigma(\cdot)$ is sigmoid function.*

*Proof.* The sigmoid function is defined as

$$\sigma(z) = \frac{1}{1 + e^{-z}}.$$

By the Mean-Value Theorem, for $z, z' \in [-R, R]$, there exists some $c$ between $z$ and $z'$ such that

$$\frac{\sigma(z) - \sigma(z')}{z - z'} = \sigma'(c),$$

where $\sigma'(c)$ is the derivative of the sigmoid function evaluated at $c$.

The derivative of the sigmoid function is

$$\sigma'(c) = \sigma(c)(1 - \sigma(c)).$$

Thus, we can rewrite the ratio as

$$\left|\frac{z - z'}{\sigma(z) - \sigma(z')}\right| = \frac{1}{\sigma'(c)} = \frac{1}{\sigma(c)(1 - \sigma(c))}.$$

Over the range $z \in [-R, R]$, the minimum value of $\sigma'(z)$ is achieved at $z = R$ or $z = -R$ with

$$\sigma'(R) = \sigma'(-R) = \frac{e^R}{(1 + e^R)^2}.$$

Thus, we have

$$\frac{1}{\sigma'(c)} \leqslant \frac{(1 + e^R)^2}{e^R} = e^{-R} + 2 + e^R.$$ □

## D. Additional Details on Section 4

In this section, we provide the proof for our main result in Section 4. As in the BT-preference model, our proof for the general preference model is modular. We first present a meta theorem of iterative Square$\chi$PO in Algorithm 3.

**Algorithm 3** Iterative `Square`$\chi$`PO` under Corruption and Privacy Protection

1: **Input:** Labeled preference dataset: locally private and corrupted $\widetilde{\mathcal{D}}_{\mathsf{pref}} = \{(x_i, a_i^0, a_i^1, z_i)\}_{i=1}^n$ under CTL and LTC, or label corrupted dataset $\bar{\mathcal{D}}_{\mathsf{pref}} = \{(x_i, a_i^0, a_i^1, \bar{y}_i)\}_{i=1}^n$ under cDP; privacy parameter $\varepsilon$; preference model class $\mathcal{L}$; policy class $\Pi$; regularization coefficient $\beta$; step size $\eta$; total number of iterations $T$

2: **Initialize:** $\pi^1 = \pi_{\mathsf{ref}}$

    // `Preference Model Estimation`

3: **if** Local model under CTL or LTC **then**

4:     Find $\widehat{\ell}$ via least-squares regression:

$$\widehat{\ell} = \underset{\ell \in \mathcal{L}}{\operatorname{argmin}} \sum_{i=1}^n \left(\ell(x_i, a_i^0, a_i^1) - c(\varepsilon)\bar{z}_i\right)^2, \tag{10}$$

    where $\bar{z}_i = 2z_i - 1$

5: **else** {Central model under cDP }

6:     Sample $\widehat{\ell}$ from $\mathcal{L}$ via the following distribution:

$$P(\ell) \propto \exp\left(-\frac{\varepsilon}{8} \cdot L(\ell; \bar{\mathcal{D}}_{\mathsf{pref}})\right),$$

    where $L(\ell; \bar{\mathcal{D}}_{\mathsf{pref}}) = \sum_{i=1}^n \left(\ell(x_i, a_i^0, a_i^1) - \bar{y}_i'\right)^2$ and $\bar{y}_i' = 2\bar{y}_i - 1$

7: **end if**

    // `Policy Optimization`

8: Collect $m$ samples $\mathcal{D}_x = \{(x, a, b)\}$, where each sample is drawn i.i.d. from $x \sim \rho$, $a \sim \pi_{\mathsf{ref}}(x)$, $b \sim \pi_{\mathsf{ref}}(x)$

9: **for** $t = 1, \dots, T$ **do**

10:     Sample $b_t \sim \pi^t(x)$ and let $\widehat{r}^t(x, a) = \widehat{\ell}(x, a, b_t)$ for all $x \in \mathcal{X}$, $a \in \mathcal{A}$

11:     Update policy by solving:

$$\pi^{t+1} = \underset{\pi \in \Pi}{\operatorname{argmin}} \mathcal{L}_t(\pi; \mathcal{D}_x),$$

    where

$$\mathcal{L}_t(\pi; \mathcal{D}_x) = \sum_{(x,a,b) \in \mathcal{D}_x} \left(\mathsf{clip}_4\left(f_{\pi,\pi^t}^{\beta,\eta}(x, a, b)\right) - \widehat{r}_{\mathsf{diff}}^t(x, a, b)\right)^2, \tag{11}$$

    with $f_{\pi,\pi^t}^{\beta,\eta}(x, a, b)$ defined in (8), and $\widehat{r}_{\mathsf{diff}}^t(x, a, b) := \widehat{r}^t(x, a) - \widehat{r}^t(x, b)$

12: **end for**

13: **Output:** $\widehat{\pi} = \mathsf{unif}(\{\pi^t\}_{t=1}^T)$

---

**Theorem D.1.** *Under the general preference model, let Assumptions 4.1, 4.2 and 4.3 hold. Then, Algorithm 3 achieves the following general duality gap across different settings:*

$$\mathsf{DG}(\widehat{\pi}) \lesssim \mathsf{subopt}(\widehat{\pi}, C) + \frac{C\beta}{\eta T} + C\beta + \frac{\eta}{\beta} + V_{\max}\sqrt{C\mathsf{err}_{\mathsf{md}}^2} + \frac{V_{\max}^2 \mathsf{err}_{\mathsf{md}}^2}{2\beta} + \frac{C\mathsf{err}_{\mathsf{general}}^2}{\beta} + \sqrt{C\mathsf{err}_{\mathsf{general}}^2} + \sqrt{\frac{\log\frac{|\Pi|}{\delta}}{T}},$$

*where* $\mathsf{subopt}(\widehat{\pi}, C) := \max_{\pi \in \Pi} \ell^*(\pi, \widehat{\pi}) - \max_{\pi \in \Pi_C} \ell^*(\pi, \widehat{\pi})$ *and* $\Pi_C := \{\pi : \max_{x \in \mathcal{X}} D_{\chi^2}(\pi(x) \| \pi_{\mathsf{ref}}(x)) \leqslant C\}$, $\mathsf{err}_{\mathsf{md}}^2 \lesssim \frac{\log(|\Pi|/\delta)}{m}$ *and* $\mathsf{err}_{\mathsf{general}}^2$ *is defined as:*

$$\mathsf{err}_{\mathsf{general}}^2 := \mathbb{E}_{x \sim \rho, a^0 \sim \pi_{\mathsf{ref}}(x), a^1 \sim \pi_{\mathsf{ref}}(x)} \left[\left(\widehat{\ell}(x, a^0, a^1) - \ell^\star(x, a^0, a^1)\right)^2\right]$$

*for the estimate* $\widehat{\ell}$ *generated by Algorithm 3 under* CTL, LTC *and* cDP.

*Proof.* This result follows from the proof of Theorem 6.2 in Huang et al. (2024). Again, our new loss will only impact the term $\mathsf{err}_{\mathsf{general}}^2$ while keeping the analysis of other parts the same. $\square$

Next, via a direct application of Lemma B.1 with a straightforward mapping in this case, we can bound the term $\text{err}^2_{\text{general}}$ under different cases, as stated in the following lemma.

**Lemma D.2.** *Under the same conditions of Theorem D.1,* $\text{err}^2_{\text{general}}$ *for Algorithm 3 satisfies the following bound with probability at least $1 - \zeta$*

$$\text{err}^2_{\text{general,CTL}} \lesssim c(\varepsilon)^2 \cdot \frac{\log(|\mathcal{L}|/\zeta)}{n} + \alpha,$$

$$\text{err}^2_{\text{general,LTC}} \lesssim c(\varepsilon)^2 \cdot \frac{\log(|\mathcal{L}|/\zeta)}{n} + \alpha \cdot c(\varepsilon),$$

$$\text{err}^2_{\text{general,cDP}} \lesssim \frac{\log(|\mathcal{L}|/\zeta)}{n} + \frac{\log(|\mathcal{L}|/\zeta)}{n\varepsilon} + \alpha\,.$$

Combining the above two results, we have the following result, which is a detailed version of Theorem 4.4 in the main body.

**Theorem D.3.** *Fix any $\zeta \in (0,1]$. Let Assumptions 4.1, 4.2 and 4.3 hold. Suppose Algorithm 3 is invoked with $\beta = \frac{1}{\sqrt{T}}$ and $\eta = \frac{1}{T}$, and for the following choices of $T$, we have with probability at least $1 - \zeta$:*

$$\text{DG}_{\text{CTL}}(\widehat{\pi}) \lesssim \min_{C \geqslant 1} \left\{ \text{subopt}(\widehat{\pi}, C) + C \left( V_{\max} \frac{\log(|\Pi|/\delta)}{\sqrt{m}} + c(\varepsilon) \sqrt{\frac{\log(|\mathcal{L}||\Pi|/\delta)}{n}} + \sqrt{\alpha \log(|\Pi|/\delta)} \right) \right\},$$

*for $T = \frac{mn}{nV^2_{\max} + m \cdot c(\varepsilon)^2 \cdot \log(|\mathcal{L}|/\zeta) + mn \cdot \alpha}$;*

$$\text{DG}_{\text{LTC}}(\widehat{\pi}) \lesssim \min_{C \geqslant 1} \left\{ \text{subopt}(\widehat{\pi}, C) + C \left( V_{\max} \frac{\log(|\Pi|/\delta)}{\sqrt{m}} + c(\varepsilon) \sqrt{\frac{\log(|\mathcal{L}||\Pi|/\delta)}{n}} + \sqrt{\alpha c(\varepsilon) \log(|\Pi|/\delta)} \right) \right\},$$

*for $T = \frac{mn}{nV^2_{\max} + m \cdot c(\varepsilon)^2 \cdot \log(|\mathcal{L}|/\zeta) + mn \cdot \alpha c(\varepsilon)}$;*

$$\text{DG}_{\text{cDP}}(\widehat{\pi}) \lesssim \min_{C \geqslant 1} \left\{ \text{subopt}(\widehat{\pi}, C) + C \left( V_{\max} \frac{\log(|\Pi|/\delta)}{\sqrt{m}} + \left( 1 + \frac{1}{\sqrt{\varepsilon}} \right) \sqrt{\frac{\log(|\mathcal{L}||\Pi|/\delta)}{n}} + \sqrt{\alpha \log(|\Pi|/\delta)} \right) \right\},$$

*for $T = \frac{mn}{nV^2_{\max} + m \cdot \left( 1 + \frac{1}{\sqrt{\varepsilon}} \right)^2 \cdot \log(|\mathcal{L}|/\zeta) + mn \cdot \alpha}$ . Furthermore, if we define the* unilateral concentrability coefficient *as*

$$C_{\text{uni}} := \max_{\pi \in \Pi, x \in \mathcal{X}, a, b \in \mathcal{A}} \frac{\pi(a \mid x) \pi_{\text{MW}}(b \mid x)}{\pi_{\text{ref}}(a \mid x) \pi_{\text{ref}}(b \mid x)},$$

*then the three bounds above imply that*

$$\text{DG}_{\text{CTL}}(\widehat{\pi}) \lesssim C_{\text{uni}} \cdot \left( V_{\max} \frac{\log(|\Pi|/\delta)}{\sqrt{m}} + c(\varepsilon) \sqrt{\frac{\log(|\mathcal{L}||\Pi|/\delta)}{n}} + \sqrt{\alpha \log(|\Pi|/\delta)} \right),$$

$$\text{DG}_{\text{LTC}}(\widehat{\pi}) \lesssim C_{\text{uni}} \cdot \left( V_{\max} \frac{\log(|\Pi|/\delta)}{\sqrt{m}} + c(\varepsilon) \sqrt{\frac{\log(|\mathcal{L}||\Pi|/\delta)}{n}} + \sqrt{\alpha c(\varepsilon) \log(|\Pi|/\delta)} \right),$$

*and*

$$\text{DG}_{\text{cDP}}(\widehat{\pi}) \lesssim C_{\text{uni}} \cdot \left( V_{\max} \frac{\log(|\Pi|/\delta)}{\sqrt{m}} + \left( 1 + \frac{1}{\sqrt{\varepsilon}} \right) \sqrt{\frac{\log(|\mathcal{L}||\Pi|/\delta)}{n}} + \sqrt{\alpha \log(|\Pi|/\delta)} \right)\,.$$

*Remark* D.4. The *unilateral concentrability coefficient* follows from the one in Cui & Du (2022), which is also used in iterative $\chi$PO (Huang et al., 2024).

# E. Experiments

**Dataset.** We utilize GPT-4o to generate a synthetic dataset, referred to as `finance_preference`, which comprises 1697 preference samples. Each sample includes a prompt related to a financial scenario and two possible responses, where "rejected" represents the high-risk option and "chosen" represents the low-risk option. This labeling can be viewed as private or sensitive information. For illustrative examples from our dataset, please refer to Appendix F. For SFT training, we construct the `finance_sft` dataset by simply concatenating the prompt with the corresponding "chosen" response.

**SFT Training.** We begin by fine-tuning GPT2-large using the `finance_sft` dataset to obtain the SFT policy, $\pi_{\text{sft}}$. For this, we directly utilize the SFT trainer from the Transformer Reinforcement Learning (TRL) library (von Werra et al., 2020).

$\chi$PO **and** Square$\chi$PO **training.** For alignment training, we split the dataset into $85\%$ for training, $5\%$ for validation, and $10\%$ for testing. For $\chi$PO, we follow the implementations in Huang et al. (2024). For Square$\chi$PO, we simply modify the log-loss to square loss as in our presented algorithm.

CTL **and** LTC **Settings.** The LDP mechanism follows the randomized response model, where the flip rate is given by $\frac{1}{e^\varepsilon+1}$. To implement both privacy and corruption, we introduce a mask variable initialized to $0$ for each sample. The LDP mechanism flips the mask variable with probability $\frac{1}{e^\varepsilon+1}$, while the corruption mechanism sets the mask to $1$ with probability $\alpha$. Finally, after CTL or LTC processing, labels ("chosen" and "rejected") are flipped if the corresponding mask value is $1$. At this point, an astute reader may notice that LTC results in a higher number of 1s in the final mask variables compared to CTL.

**Evaluation.** Evaluation. We evaluate our trained models by generating responses for the test dataset. To assess performance, we employ the `llama3:70b` model as a judge, comparing responses from $\chi$PO and Square$\chi$PO PO against those from $\pi_{\text{sft}}$. Finally, we use the win rate from these comparisons as our primary performance metric. We compute the average and standard deviation across 5 random seeds.

**Results.** We have compared the performance of $\chi$PO and Square$\chi$PO under CTL and LTC settings with $\varepsilon = 0.5$ and $\alpha = 0.1$. In particular, the following table gives the win rate ($\%$) over the $\pi_{\text{sft}}$ for different settings. We can see that (i) there exists a separation between LTC and CTL, and (ii) our Square$\chi$PO outperforms $\chi$PO in both settings.

| Setting | $\chi$PO | Square$\chi$PO |
|---------|----------|----------------|
| CTL | $64.2 \pm 0.03$ | $67.0 \pm 0.05$ |
| LTC | $59.8 \pm 0.02$ | $60.0 \pm 0.02$ |

*Table 1.* Performance comparison of $\chi$PO and Square$\chi$PO under CTL and LTC settings.

# F. Additional Details on Experiments

Below, we present a selection of examples from our generated financial dataset across various categories. Each example demonstrates a prompt alongside "Chosen" and "Rejected" responses, illustrating the alignment of decisions with risk levels and priorities.

**Category: Lifestyle & Personal Planning**
**Prompt:** "You're saving \$3,000 to host a family talent show. How do you proceed?"
**Chosen:** "Rent a small venue and create DIY props and prizes."
**Rejected:** "Spend on professional staging and lighting for a one-time event."

**Category: Home Improvement & Maintenance**
**Prompt:** "You're saving \$10,000 to add an outdoor kitchen. How do you proceed?"
**Chosen:** "Install a grill, sink, and storage with weather-resistant materials."
**Rejected:** "Spend on high-end appliances that exceed your budget."

**Category: Investments**
**Prompt:** "You're saving \$12,500 to invest in green construction funds. How do you proceed?"
**Chosen:** "Choose funds with diverse holdings in sustainable building materials."

**Rejected:** "Invest in speculative green startups with limited financial history."

**Category: Small Business Ventures**
**Prompt:** "You're saving $10,000 to start a custom clothing line. How do you proceed?"
**Chosen:** "Focus on affordable designs and use an online platform to sell."
**Rejected:** "Spend on a luxury boutique storefront before establishing demand."

**Category: Education & Skill Development**
**Prompt:** "You're saving $5,000 to attend a data visualization course. How do you proceed?"
**Chosen:** "Enroll in a course with interactive projects and industry relevance."
**Rejected:** "Choose a program with limited hands-on training."

**Category: Debt Management**
**Prompt:** "You're saving $12,000 to pay off a business loan. How do you proceed?"
**Chosen:** "Apply the funds directly to reduce the principal and future interest."
**Rejected:** "Use the funds for operational expenses while extending the loan term."

**Category: Miscellaneous**
**Prompt:** "You want to save $4,500 to organize a youth art festival. How do you proceed?"
**Chosen:** "Partner with local sponsors and focus on cost-effective exhibits."
**Rejected:** "Spend heavily on promotional campaigns without engaging artists."

These examples illustrate the structured nature of our dataset and its alignment with decision-making scenarios across diverse financial categories.

