# OpenReview forum: "Square$\chi$PO: Differentially Private and Robust $\chi^2$-Preference Optimization in Offline Direct Alignment"
_ICML.cc/2025/Conference — ICML 2025 poster_

### Official Review · Reviewer_G75h · 2025-03-13

**Overall Recommendation:** 3

**Summary:**

This paper studies the problem of alignment of language models with preference feedback, under two variations: (i) label corruption and (ii) privacy protections. While motivated by language models, there is nothing is specific to language models in the techniques, and they are more generally applicable to any offline alignment problem.

In the offline alignment problem, we are given a dataset where each "example" contains a tuple $(x, a^0, a^1, y)$ where,
* $x$ is drawn from some distribution $\rho$,
* $a^0$ and $a^1$ are two independent draws from some reference policy $\\pi\_{ref}(\\cdot | x)$, and
* $y \\in \\{0, 1\\}$ indicates the preference between $a^0$ and $a^1$, that is sampled from the Bernoulli distribution $\\mathrm{Ber}({\\cal P}^*(a^1 > a^0 | x))$.

The goal is to learn a "good policy" $\\pi(\\cdot | x)$. There are two ways considered for quantifying how good a policy is.

1. In the Bradley-Terry preference model, it is assumed that ${\\cal P}^*(a^1 > a^0 | x)$ is defined as $\\frac{e^{r^*(x, a^1)}}{e^{r^*(x, a^1)} + e^{r^*(x, a^0)}}$ for some reward function $r^*(x, a)$, and the goal is to learn a policy $\\widehat{\\pi}$ that minimizes the suboptimality gap $J(\\pi^*) - J(\\widehat{\\pi})$ where $J(\\pi) := \\mathbb{E}_{x \\sim \\rho, a \\sim \\pi(\\cdot | x)} r^*(x, a)$.

2. In the General Preference model, there is no assumed parameterization of $({\\cal P}^*(a^1 > a^0 | x)$. Here the quality of a policy $\\widehat{\\pi}$ is measured in terms of a duality gap (see the paper for definition).

Two models of data perturbation are considered:
1. Label corruption (in particular, a Huber model of corruption of the label $y$), and Local differential privacy (where the label $y$ is randomized with some known probability). These are considered in both orders (first Label corruption then Local DP or vice versa).

2. Label corruption and central DP, wherein, the labels are assumed to be corrupted as per the Huber model, and then some central differentially private mechanism is applied for learning.

The paper proposes a new policy learning algorithm referred to as _Square$\\chi$PO_. For both the settings of data perturbation, the paper provides provable upper bounds on the quality of the learnt policy. (For pertubration of type 1, the rates are shown for both label corruption then local DP, and vice versa).

### Post-rebuttal update

I thank the authors for the discussion, and I will maintain my score.

**Claims And Evidence:**

All claims are supported by proofs in the Appendix.

**Essential References Not Discussed:**

As far as I can tell, all essential references have been adequately discussed.

**Experimental Designs Or Analyses:**

There are no experiments in the paper, so this question is not relevant.

**Methods And Evaluation Criteria:**

There are no experiments in the paper, so this question is not relevant.

**Other Comments Or Suggestions:**

I think it would be beneficial to discuss some motivation for when $\\mathsf{CTL}$ and $\\mathsf{LTC}$ settings could be applicable in practice.

I can see why $\\mathsf{CTL}$ makes sense: the corruption is a way to handle misspecification in the model, and the Local DP randomization is added for privacy. But I don't know where $\\mathsf{LTC}$ comes up.

**Other Strengths And Weaknesses:**

The paper considers a nice twist on the offline alignment problem of label corruption and differential privacy, and provides upper bounds on the sub-optimality gap / duality gap.

One thing that was not clear to me was how tight the established bounds are. Without the tightness results, it is difficult to be convinced of the exact interplay between label corruption and DP guarantees.

**Questions For Authors:**

Is it clear how tight the established upper bounds on sub-optimality gap and duality gap are? Is it possible to prove any lower bounds?

**Relation To Broader Scientific Literature:**

The paper studies the offline alignment problem in the context of label corruption and differential privacy. This seems novel and interesting to me.

**Theoretical Claims:**

I looked at the theoretical claims at a high level, and only briefly skimmed the Appendix.

---

> ### Author Rebuttal · Authors · 2025-03-30
>
> We thank the reviewer for the detailed feedback and constructive suggestions. We address the main points below and hope it will help to resolve your concerns.
>
> **1. Motivation of LTC.** The main motivation behind LTC is that after users privatize their preferences, the collected preference signals may be corrupted—either due to communication errors or adversarial attacks in the collection/transmission process.
>
> **2. Tightness of bounds.** Thanks for this sharp question. We provide our thoughts in detail below.
>
> - **Privacy only.** For the local model under BT-preference, when there is no corruption ($\alpha = 0$), our rate is minimax optimal [R1]. Under the central model, we conjecture that the current $1/\sqrt{n \epsilon}$ additive cost cannot be improved if one relies on the statistical error in $L_2$. One possible way to improve this is to work with the $L_1$ norm, which we believe is a promising direction for achieving a better (and potentially optimal) $1/(n\epsilon)$ additive privacy cost.
>
> - **Corruption only.** When there is no privacy constraint, our current rate is $O(\sqrt{\alpha})$ under Huber corruption. We conjecture this to be suboptimal compared to the ideal $O(\alpha)$ rate. However, our current rate matches existing results in [R2], which considers standard offline RL with direct observation of rewards.
>
> - **Interplay between privacy and corruption.** Currently, under LTC, our rate is approximately $O(\sqrt{\alpha / \epsilon})$ when $\epsilon \le 1$. This differs from the known interplay in the mean estimation problem, which yields a rate of $O(\alpha/\epsilon)$ [R3]. We leave a careful study of the precise interaction between privacy and corruption as an interesting direction for future work.
>
> ---
>
> [R1] Chowdhury, S. R., Zhou, X., and Natarajan, N. Differentially private reward estimation with preference feedback.
> In International Conference on Artificial Intelligence and
> Statistics, pp. 4843–4851. PMLR, 2024b.
>
> [R2] Zhang, X., Chen, Y., Zhu, X., and Sun, W. Corruption-robust offline reinforcement learning. In International
> Conference on Artificial Intelligence and Statistics, pp.
> 5757–5773. PMLR, 2022.
>
> [R3] Zhou, X. and Zhang, W. Locally private and robust multiarmed bandits. In The Thirty-eighth Annual Conference
> on Neural Information Processing Systems, 2024.

---

### Official Review · Reviewer_Rnup · 2025-03-13

**Overall Recommendation:** 4

**Summary:**

The paper studies algorithms for alignment given privacy and robustness considerations. In alignment we are given examples x, two model responses $a_0, a_1$, and a label $y$ denoting that $a_y$ was preferred to $a_{1-y}$. The labels are generated under one of two models, a reward model where each label-response pair has a reward and $y$ is set to 0/1 w.p. proportional to exponential in the reward of the corresponding action, or a generalized model where each pair can have arbitrary probability of preference, and preferences need not be transitive. With privacy, the label is privatized using randomized response (i.e. flipped w.p. $1 / (1 + e^\epsilon)$ for some $\epsilon$), and under corruption the true distribution is replaced with an unknown Bernoulli w.p. $\alpha \leq 1/2$. Given a policy class, the canonical DPO picks a policy that maximizes the sum of the log a certain utility function over the data. $\chi$PO is a recent modification of DPO that adds a regularization term to the utility function. The authors propose Square$\chi$PO, which minimizes a squared loss on the $\chi$PO utility function instead of the log. Square$\chi$PO also uses a scaling to de-bias the labels after randomized response. For preferences determined by a reward function, under a single-policy concentrability assumptioned used in $\chi$PO, the authors bound the suboptimality of Square$\chi$PO. The suboptimality bound consists of a bias term $\sqrt{\alpha}$ (that is $c(\epsilon)$ larger if the labels are corrupted after instead of before privatization, where $c(\epsilon)$ is the scaling for de-biasing) plus a term $c(\epsilon) \sqrt{(\log |\Pi|)/n}$, where $\Pi$ is the size of the policy class. This retrieves the $1/\sqrt{n}$ optimal dependence on dataset size. The authors also consider a central model where the entire example (not just the label) is private, but does not need to be privatized with local DP. They propose an exponential mechanism based on their square loss, and show similar guarantees. They also show similar guarantees for general preference models instead of preferences dictated by a reward function.

## update after rebuttal
I remain in support of accepting the paper

**Claims And Evidence:**

Yes

**Essential References Not Discussed:**

N/A

**Experimental Designs Or Analyses:**

N/A; there are no empirical results in the paper

**Methods And Evaluation Criteria:**

N/A; there are no empirical results in the paper

**Other Comments Or Suggestions:**

N/A

**Other Strengths And Weaknesses:**

Strengths:
* In addition to allowing both privacy and robustness, extending to general reward functions and the general preference model is a strong contribution, as many past works assume a very restrictive linear reward model (not even a general reward model). I view the primary strength of this work as its wide generality in the scope of results compared to many of the past results.
* Paper structure and presentation is quite clean, it is easy to understand the problem setup, the algorithms, and the comparisons to past work.
* Authors are transparent about the limitations of results: e.g., they achieve different results for LTC and CTL (ordering of privatization and corruption), but clarify this is evidence for added difficulty of LTC but not a formal hardness result separating the two, and they state their exponential mechanism is not feasible to run in practice.
* The algorithm and its analysis are not a trivial modification/combination of past results

Weakness:
* As the authors mention, the central DP algorithm is an exponential mechanism which may be infeasible to run in practice. However, there are no computationally efficient results for pure-central-DP private and robust alignment to compare to.

**Questions For Authors:**

No questions that would substantially affect my evaluation.

**Relation To Broader Scientific Literature:**

For local DP, a prior work of Chowdhury et al. studies robust but not private alignment, and achieves worse dependence on dataset size $1/n^{1/4}$ under the assumption that the reward is a linear model, with a stronger concentrability assumption, and does not extend to general reward functions. A different paper by Chowdhury et al. studies private but not robust alignment and only focuses on linear rewards, whereas the authors extend to general rewards. For central DP, concurrent works study weaker approximate DP while this work studies pure DP, but these works get better dimension-dependence in part due to the weaker DP definition. These works also focus on linear reward models while the present work tolerates general reward models.

**Theoretical Claims:**

Theoretical claims were not checked in detail

---

> ### Author Rebuttal · Authors · 2025-03-30
>
> Thank you for your positive evaluation of our paper. We appreciate your recognition of our general analysis with non-trivial modifications of previous results.
>
> We are also grateful for your recognition of our transparency in acknowledging the computational efficiency limitations in the central model. We'd like to take this opportunity to elaborate further. Our generic analytical framework is modular and allows for the integration of any advances in computationally efficient private regression. Under the realizability assumption, the estimation error under the square loss used in our current analysis aligns with the population excess risk in private stochastic optimization, both convex and non-convex. This means that, rather than relying on exponential mechanisms, one could substitute existing efficient methods.
> However, a key limitation is that these methods typically yield a slower $1/\sqrt{n}$ rate in the non-private term without additional structural assumptions (rather than our $1/n$ under the exponential mechanism), which ultimately translates to a worse rate of $1/n^{1/4}$ in the end (for the non-private term). Exploring how to leverage additional structure (e.g., strong convexity or even weaker assumptions) to achieve optimal rates in a computationally efficient manner is an exciting direction for future work.

---

### Official Review · Reviewer_6UpT · 2025-03-14

**Overall Recommendation:** 3

**Summary:**

The paper proposes differentially private and robust offline preference alignment with human feedback. The method is based on the prior work of $\chi$PO, but uses square loss instead of the log loss.

**Claims And Evidence:**

The paper claims to achieve optimal rates in general function approximations under privacy constraints and achieves both privacy and robustness. However, it is not clear how these are achieved by replacing log loss with square loss. Does $\chi$PO not obtain privacy and robustness?

**Essential References Not Discussed:**

I'm not aware of any related works in this area.

**Experimental Designs Or Analyses:**

The paper lacks experiments.

**Methods And Evaluation Criteria:**

There are no experimental evaluations in this paper.

**Other Comments Or Suggestions:**

It would be helpful to include an experimental evaluation of the proposed Square$\chi$PO method and compare it with prior works.

[EDIT:] I'm raising my score based on the rebuttal response. I would recommend the authors to include the experiment results discussed in the rebuttal.

**Other Strengths And Weaknesses:**

- The novelty of the approach is limited as the proposed approach is a modification of the existing $\chi$PO approach with the log loss being replaced with squared loss. While this does allow for some interesting observations for privacy and robustness alignment, it is not clear how significant the differences are from the prior works (mainly w.r.t. $\chi$PO).
- Lack of experimental comparison with other policy optimization algorithms. While the paper mainly focuses on theoretical results, their method can be implemented in practice as the authors point out. Thus, a thorough comparison with prior approaches would be useful, especially given that the method is a modification of a prior approach.

**Questions For Authors:**

Can you please highlight the key differences between $\chi$PO and Square$\chi$PO and how these are benefited by going from log loss to square loss?

**Relation To Broader Scientific Literature:**

The paper directly builds upon the previous state-of-art offline preference alignment work, $\chi$PO, that solved the overoptimization problem in direct alignment. The proposed work claims to achieve both privacy and robustness.

**Theoretical Claims:**

I did not check the theoretical proofs in the appendix.

---

> ### Author Rebuttal · Authors · 2025-03-30
>
> Thank you for time and feedback. We will recap your comments and present our detailed response. We hope our answers will resolve your concerns.
>
> **1. Significance of the difference compared to $\chi$PO**
>
> We clarify that the key significance and benefits of moving from log-loss to square loss in our Square$\chi$PO is that it allows us to address privacy and robustness **simultaneously** for **both** BT and the general preference model. The reasons behind this have been highlighted in Section 3.1.1. To recap, the boundedness of square loss (rather than the unboundedness in log-loss) allows us to handle corruption easily. Meanwhile, square loss enables us to handle BT and the general preference model in a unified manner.
>
> **2. Preliminary experiments**
>
> While our contribution is mainly a theoretical one, we have now also made an effort to run some preliminary experiments as a proof of concept.
>
> - **A quick summary of results.** We have compared the performance of $\chi$PO and Square$\chi$PO under CTL and LTC settings with $\epsilon = 0.5$ and $\alpha = 0.1$. In particular, the following table gives the win rate (%) over the `π_sft` for different settings.  We can see that (i) there exists separation between LTC and CTL, and (ii) our Square$\chi$PO
> outperforms $\chi$PO in both settings.
>
>
> | Setting | $\chi$PO          | Square$\chi$PO   |
> |:-------:|:-----------------:|:----------------:|
> | CTL     | $64.2 \pm 0.03$    | $67.0 \pm 0.05$   |
> | LTC     | $59.8 \pm 0.02$    | $60.0 \pm 0.02$   |
>
> ---
> More details about our experiments are given below:
>
>  - **Dataset.** We utilize `GPT-4o` to generate a synthetic dataset, referred to as `finance_preference`, which comprises $1697$ preference samples. Each sample includes a prompt related to a financial scenario and two possible responses, where `rejected` represents the high-risk option and `chosen` represents the low-risk option. This labeling can be viewed as private or sensitive information. For SFT training, we construct the `finance_sft` dataset by simply concatenating the prompt with the corresponding `chosen` response.
>
>  - **SFT Training.**  We begin by fine-tuning `GPT2-large` using the `finance_sft` dataset to obtain the SFT policy, `π_sft`. For this, we directly utilize the SFT trainer from the Transformer Reinforcement Learning (`TRL`) library
>
>  - **$\chi$PO and Square$\chi$PO Training.**  For alignment training, we split the dataset into `85%` for training, `5%` for validation, and `10%` for testing.  For $\chi$PO, we follow the implementations in Huang et al. For Square$\chi$PO, we simply modify the log-loss to square loss as in our presented algorithm.
>
>  - **CTL and LTC Settings.**  The LDP mechanism follows the randomized response model, where the flip rate is given by $1 / (e^ε + 1)$. To implement both privacy and corruption, we introduce a mask variable initialized to `0` for each sample. The LDP mechanism flips the mask variable with probability $1 / (e^ε + 1)$, while the corruption mechanism sets the mask to `1` with probability `α` (different from random flipping).  Finally, after CTL or LTC processing, labels (`chosen` and `rejected`) are flipped if the corresponding mask value is `1`.
>
>  - **Evaluation.**  We evaluate our trained models by generating responses for the test dataset. To assess performance, we employ the `llama3:70b` model as a judge, comparing responses from $\chi$PO and Square$\chi$PO against those from `π_sft`.  Finally, we use the win rate from these comparisons as our primary performance metric. We compute the average and standard deviation across `5` random seeds.

---

### Official Review · Reviewer_bvkr · 2025-03-16

**Overall Recommendation:** 4

**Summary:**

LLMs is important to have alignment with human response. This work focuses on an approach from direct preference optimization, especially on CHI-PO. This approach is to address overoptimization issue in DPO on single-policy concentrability, which is a kind of offline alignment approach. In such approach, privacy and robustness of preference datasets are not well studied.

On privacy part, this work proposes Square CHI-PO by using a new square loss over probabilities in vanilla CHI-PO with differential privacy for general function approximations. On robustness part, Square CHI-PO preserves vanilla CHI-PO’s single-policy concentrability. It also achieves optimal rate for the approximation against random-flipping corruption as well as Huber label corruption.

**Claims And Evidence:**

The claims is well stated the gap of offline DPO by observing limited work on theoretical guarantees for both privacy and robustness. None current work is sufficient to guarantee general function approximation. In advance, this work applies a square-based loss with DP to replace the log-based loss in CHI-PO from Huang et al., 2024 to balance both privacy and robustness.

**Essential References Not Discussed:**

To my best knowledge, this work has discussed the most essential related work for its topic.

**Experimental Designs Or Analyses:**

Although the theoretical claims and analysis are good, this work is lack of experimental results to support the theory, so it is a pure theory paper.

**Methods And Evaluation Criteria:**

Algorithm 1 and 2 well sketch the procedure to make this work tackle the issue. However, since this work is a pure theory work, authors did not provide any evaluation criteria and experimental results.

**Other Comments Or Suggestions:**

N/A

**Other Strengths And Weaknesses:**

N/A

**Questions For Authors:**

N/A

**Relation To Broader Scientific Literature:**

This work has well discussed DPO work from Rafailov et al., 2023 and the most related variant CHI-PO from Huang et al. 2024 on either privacy or robustness perspective in the introduction.

(1) For privacy, Chowdhury et al., 2024b and Korkmaz & Brown-Cohen, 2024 work on linear function approximation and are insufficient for non-linear or policy functions, while this work works for general function approximation.

(2) For robustness, Mandal et al. 2024 uses RLHF-based method for linear setting, and Chowdhury et al., 2024a uses DPO approach, but with a suboptimal rate. None of them tackles the robustness for general function approximation. For Huber label corruption, this work matches offline RL setting in Zhang et al., 2022.

In addition, in preliminaries, this work discuss two offline alignment approaches, where Bradley & Terry, 1952 is to learn a policy to minimize the suboptimality gap, and Munos et al., 2023 captures non-transitive preferences without using reward function, which relies on minimax function.

**Theoretical Claims:**

Section 3 and 4 as well as appendix Section C and D provides enough details to show the effectiveness of this work in the theoretical perspective.

---

> ### Author Rebuttal · Authors · 2025-03-30
>
> Thank you for your positive evaluation of our paper. We appreciate your recognition of our theoretical contributions to offline alignment, as well as our approach to privacy and robustness.
>
> We're also grateful for your appreciation of a purely theoretical paper. To further support our main results, we have made an effort to include some preliminary empirical results as a proof of concept. Please see our response to Reviewer 6UpT.

---

### Decision · Program_Chairs · 2025-05-01

**Decision:**

Accept (poster)

**Comment:**

This paper proposed new algorithms for private and robust alignment of language models by modifying a previous algorithm CHI-PO. This algorithm enables nice theoretical gurantees and most of the reviewers found the theoretical contribution valuable. The authors also provided preliminary empirical studies during the rebuttal which further demonstrated the efficacy of the proposed method.